# Generalization in LLM Problem Solving: The Case of the Shortest Path

**Yao Tong**[1]    **Jiayuan Ye**[1]    **Anastasia Borovykh**[2]    **Reza Shokri**[1,3]
[1]National University of Singapore    [2]Capital Fund Management    [3]Google Research
[1]{yaotong, jiayuan, reza}@comp.nus.edu.sg
[2]anastasia.borovykh@cfm.com

## Abstract

Whether language models can systematically generalize remains actively debated. Yet empirical performance is jointly shaped by multiple factors such as training data, training paradigms, and inference-time strategies, making failures difficult to interpret. We introduce a controlled synthetic environment based on shortest-path planning, a canonical composable sequential optimization problem. The setup enables clean separation of these factors and supports two orthogonal axes of generalization: *spatial transfer* to unseen maps and *length scaling* to longer-horizon problems. We find that models exhibit strong spatial transfer but consistently fail under length scaling due to recursive instability. We further analyze how distinct stages of the learning pipeline influence systematic problem-solving: for example, data coverage sets capability limits; reinforcement learning improves training stability but does not expand those limits; and inference-time scaling enhances performance but cannot rescue length-scaling failures. Code is available at https://github.com/privacytrustlab/PathGeneralization.

## 1    Introduction

Despite rapid progress in post-training methods (Tie et al., 2025; Guo et al., 2025; Yao et al., 2023; Peng et al., 2023; Wei et al., 2021) and reasoning benchmarks (Balunović et al., 2025; Rein et al., 2024; Jain et al., 2024; Jimenez et al., 2024), it remains unclear and actively debated (Zhao et al., 2024; Lampinen et al., 2025; Xu et al., 2024; Liu et al., 2025b; Xu et al., 2024; Ramesh et al., 2023; Dziri et al., 2023) whether language models can systematically generalize.

A central challenge is that performance on reasoning benchmarks is jointly influenced by multiple factors, including training and testing data properties (Wold et al., 2025; Chang et al., 2025), training paradigms such as supervised fine-tuning (SFT) or reinforcement learning (RL) (Yue et al., 2025b; Wu et al.; Liu et al., 2025b;a), and inference-time strategies (Wang et al., 2022; Li et al., 2023b; Mudgal et al., 2023). Consequently, observed reasoning failures may arise from insufficient data coverage, training dynamics that fail to induce the underlying optimization rules, or inference procedures that cannot effectively express capabilities already present in the model. Moreover, natural benchmarks often make it difficult to determine whether a model truly generalizes, as the differences between training and evaluation settings are not cleanly controlled. For instance, it is often unclear whether test tasks require genuinely new skills or can still be solved using patterns observed during training, and whether the training and test distributions are truly disjoint.

To address these challenges, we construct a controlled synthetic environment grounded in a well-defined class of problems: composable *sequential optimization problems* (SOPs). Many structured reasoning and planning tasks (e.g., problems solvable via dynamic programming) can be viewed as composable SOPs, where solving a task requires composing a sequence of locally valid decisions, each influencing subsequent decisions, such that the entire sequence satisfies a global objective. Among such tasks, shortest-path planning provides a canonical instantiation with a globally verifiable objective and unambiguous optimal solutions. We evaluate models in a direct-answer setting, requiring them to generate the complete solution path given start and end nodes without intermediate reasoning steps. Based on this controlled synthetic environment, we ask three related questions:

**(1) Can language models systematically generalize on composable SOPs?**

**(2) If a model can solve smaller instances, can it compose them to solve larger-scale or structurally novel problems?**

**(3) How is such generalization capability shaped across different stages of the learning pipeline**, including training data, training paradigms, and inference-time scaling methods?

This environment naturally supports two axes of out-of-distribution generalization evaluation: **spatial transfer**, which tests whether the model can solve the task in entirely unseen maps, and **length scaling**, which evaluates the ability to handle longer-horizon problems within the same map. *Success under these settings provides a direct operational test of whether the model systematically generalizes*. This controlled setup makes it possible to independently vary (i) training data properties, (ii) training paradigms, including SFT and RL (enabled by the easily verifiable reward), and (iii) inference-time strategies. This separation enables us to isolate how different components of the learning pipeline shape distinct dimensions of generalization in SOPs. The structured map environment further supports direct analysis and visualization of characteristic failure modes, offering insight into the model's underlying behavior.

Through extensive controlled experiments, we obtain the following key findings:

1. Models can spatially transfer to entirely unseen maps, providing evidence of systematic structural generalization, but fail to generalize under length scaling, primarily due to recursive instability (Section 3).
2. Training data primarily determines capability limits: allocating training budget to more distinct questions rather than more solutions yields better transfer generalization. Selecting questions with broader primitive coverage is more beneficial than more diverse primitive combinations. (Section 4) Length scaling requires exposure to slightly longer training examples. (Section 5)
3. Reinforcement learning stabilizes training but does not surpass the performance ceiling of supervised fine-tuning (SFT), and exhibits error patterns similar to SFT (Section 6). RL remains below the best SFT performance even when stronger inference strategies are used to better unlock intrinsic capability, and appears to restrict the effective solution space (Section 7).
4. Advanced inference-time strategies can improve performance for both SFT and RL models but cannot rescue length scaling failures. (Section 7)

We defer detailed discussions of research gaps and motivations to the beginning of each section, and related work can be found in Section C.

## 2 PRELIMINARIES AND EXPERIMENTAL SETUP

**Sequential Optimization Problems.** An SOP is specified by a state space $\mathcal{S}$, an action space $\mathcal{A}$, a transition rule $T$, a goal set $\mathcal{G}$, and a global objective $\mathcal{C}$.

Given an initial state $s_0$, the goal is to find an action sequence $a_{1:T} = (a_1, \ldots, a_T)$ solving

$$\min_{a_{1:T}, T \geq 1} \mathcal{C}(s_0, a_{1:T}) \quad \text{s.t.} \begin{cases} s_{t+1} = T(s_t, a_t), \\ s_T \in \mathcal{G}. \end{cases} \tag{1}$$

We say that an SOP is *composable* if, for any intermediate state $s_k$ along an optimal trajectory, $\mathrm{Opt}(s_i, s_j) = \mathrm{Opt}(s_i, s_k) \circ \mathrm{Opt}(s_k, s_j)$, where $\circ$ denotes concatenation of two valid subtrajectories. For example, in shortest-path planning, $\mathrm{ShortestPath}(i, j) = \mathrm{ShortestPath}(i, k) \circ \mathrm{ShortestPath}(k, j)$.

**Spatial Transfer.** Our notion of structural transfer aligns with established definitions of systematic generalization (Wiedemer et al., 2023b; Fu et al., 2024), where generalization is characterized as applying known rules to novel combinations of primitives outside the training support. Formally, let $G = (V, A)$ be a *sparse grid map* (i.e., with edges blocked) with node set $V$ and adjacency $A$. A mobility *rule* $f(i, j \mid G)$ returns a mobility path from node $i$ to node $j$ under $G$. The *training support* is the set of ordered start–end node pairs used in training, $\mathrm{supp}(\mathcal{D}_{\text{train}}) \subseteq V \times V \setminus \{(i, i)\}$. We **evaluate transfer generalization of a model $\theta$ trained on $\mathcal{D}_{\text{train}}$ as its performance in applying rule $f$ to novel ordered pairs** $(i, j) \sim \mathcal{D}_{\text{test}}$, where all node pairs in the test set are disjoint from those in training, i.e., $\mathrm{supp}(\mathcal{D}_{\text{test}}) \cap \mathrm{supp}(\mathcal{D}_{\text{train}}) = \emptyset$. In our case, $\mathcal{D}_{\text{test}}$ is drawn from a disjoint novel map $\hat{G} = (\hat{V}, \hat{A})$ with $\hat{V} \cap V = \emptyset$ and $\hat{A} \neq A$, i.e., irrelevant to $G$ in nodes, edges, sparsity or size.

Such a truly disjoint test space is rarely achievable in natural language, where systematicity is often evaluated by holding out primitives within the same domain. This can yield overly optimistic estimates, since semantically similar primitives (e.g., "run" vs. "walk") may lie close in embedding space. Our spatial setup therefore provides a more faithful measure of systematic generalization.

**Length scaling.** Similarly, we formalize horizon scaling following prior definitions of length generalization (Sinha et al., 2024; Cai et al., 2025). Within the same notation, it can be viewed as a constrained form of transfer, where novelty is enforced along the path-length axis. Let $l(\mathcal{D})$ denote the set of path lengths for the mobility pairs in dataset $\mathcal{D}$. Then, in addition to the disjointness condition $\mathrm{supp}(\mathcal{D}_{\text{test}}) \cap \mathrm{supp}(\mathcal{D}_{\text{train}}) = \emptyset$, length scaling further requires $\max l(\mathcal{D}_{\text{train}}) \leq \min l(\mathcal{D}_{\text{test}})$, i.e., all test pairs must involve strictly longer paths than any seen in training.

**Metric.** Let $\hat{f}_\theta(i, j \mid G)$ denote the path predicted by the model $\theta$. We measure generalization performance using the *success rate (SR)*:

$$\text{SR} = \Pr_{(i,j)\sim\mathcal{D}_{\text{test}}} \left[ \hat{f}_\theta(i, j \mid G) = f(i, j \mid G) \right] \tag{2}$$

In our experiments, we adopt the shortest-path rule for $f$, which makes path length precisely controllable.[1] Our goal is to study the properties of the data and training paradigm rather than the inherent learnability of the task itself, and shortest-path is a canonical path-finding problem that is (theoretically) regarded learnable by language models (Cohen et al., 2025; Dai et al., 2024). In shortest-path, $f(i, j \mid G)$ may return a *set of valid paths* whenever multiple paths exist between $i$ and $j$. During evaluation in Equation (2), we deem $\hat{f}_\theta(i, j \mid G)$ successful if it belongs to the set $f(i, j \mid G)$.

**Empirical Setup.** We trained 8-layer, 8-head Transformer models from scratch following the LLaMA architecture (AI@Meta, 2024), which employs Rotary Positional Embeddings (RoPE) (Su et al., 2021) for position encoding. The models were pretrained on random-walk paths over all maps ($G$ and $\hat{G}$), simulating the autoregressive pretraining phase of large language models (LLMs). This pretraining exposes the model to the primitives (nodes) and their semantics, defined by their adjacency relationships. To prevent interference with downstream mobility learning tasks, we bias the pretraining distribution by constraining random-walk paths to have a minimum length substantially longer than any path in the fine-tuning distribution. (We also validate this non-interference in Section D.3.) This mirrors common practice in LLM pretraining, where models are exposed to much longer sequences than those used in fine-tuning or evaluation. Further pretraining details are provided in Section D.1.

For evaluation, we fine-tune the models on shortest paths on the training map $G = (V, A)$. We split the node set $V$ into training and test sets: the training sets contains 80% of the nodes (from which a subset of nodes $V_{\text{train}}$ used to form $\mathcal{D}_{\text{train}}$ is sampled) and the remaining 20% for length scaling evaluation. We test spatial transfer on different disjoint test maps $\hat{G} = (\hat{V}, \hat{A})$.

**Training paradigms and data format.** We study two training paradigms: supervised fine-tuning (SFT) and reinforcement learning (RL). For SFT, each training sample is represented as a sequence of the form ` i j : i E S E E ... N E S W W j `, where `i` and `j` denote the start and end nodes, `` and `` are special tokens, and the path is encoded as a sequence of movement directions (E, W, N, S). Using directions instead of node indices prevents the model from trivially memorizing n-gram sequences of node identifiers. The prompt prefix ` i j :`, which we refer to as the *question* such that the path itself is the *answer*, is excluded from the loss during SFT. At test time, we feed this prompt to the model and evaluate the generated continuation, i.e., asking the question "what is the shortest path from $i$ to $j$?".

Our path setup also naturally lends itself to RL for two reasons. (1) The shortest paths are inherently verifiable: a generated sequence either forms a valid shortest path or not, allowing us to define a binary reward without additional heuristics; (2) Although the model is not explicitly designed to "think", the path-generation process itself resembles a step-by-step reasoning procedure, making RL a natural training paradigm for this setting. We adopt the Dr.GRPO (Liu et al., 2025c) algorithm (an unbiased variant of GRPO and the de facto standard in recent implementations of RL with LLMs), with a binary reward of 1 if the generated sequence forms a valid shortest path between `i` and `j` and 0 otherwise. The RL-trained model is trained on the same prompt prefix ` i j :`, and we vary the number of rollouts per prompt (4, 8, and 16) during training.

---

[1]Many other common mobility rules, such as DFS, yield unconstrained lengths.

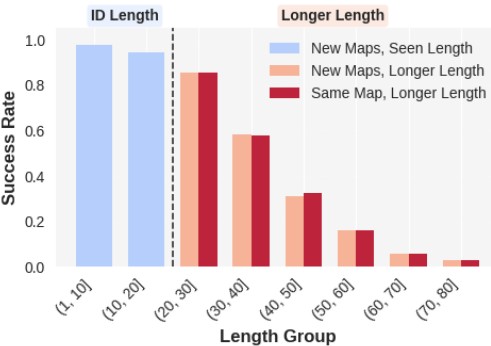

Figure 1: The model successfully transfers to unseen test maps within the training length but fails to generalize to longer paths. The vertical dashed line denotes the boundary between training and longer lengths.

| Metrics | Length groups | (20,30] | (30,40] |
|---|---|---|---|
| $\Pr(\text{Long})$ | | 0.774 | 0.530 |
| $\Pr(\text{Sub})$ | | 0.920 | 0.893 |
| $\Pr(\text{Sub}_1 \wedge \text{Sub}_2)$ | | 0.846 | 0.796 |
| $\Pr(\text{Long} \mid \text{Sub}_1 \wedge \text{Sub}_2)$ | | **0.811** | **0.589** |
| $\Pr(\text{Long}, \neg(\text{Sub}_1 \wedge \text{Sub}_2))$ | | 0.082 | 0.061 |

Table 1: Composition analysis of length scaling failure. The full-path success rate $\Pr(\text{Long})$ can be decomposed into $\Pr(\text{Long}, \text{Sub}_1 \wedge \text{Sub}_2)$ and $\Pr(\text{Long}, \neg(\text{Sub}_1 \wedge \text{Sub}_2))$. The degradation is primarily driven by the drop in $\Pr(\text{Long} \mid \text{Sub}_1 \wedge \text{Sub}_2)$, indicating reduced stability under recursive application of learned rules.

## 3 CAN LANGUAGE MODELS SYSTEMATICALLY GENERALIZE?

A model exhibits systematic generalization if it can apply learned rules compositionally beyond the specific configurations observed during training. This includes both solving problems built from disjoint primitives and solving problems that require deeper recursive application of the same rule. In our setting, this corresponds to generating valid shortest paths in previously unseen maps (spatial transfer) and generating valid shortest paths of greater lengths than those seen during training (length scaling). We begin by presenting the high-level generalization behavior under the strongest training configuration (Figure 1), and defer a detailed breakdown of training data and paradigm choices to later sections. Specifically, the model is evaluated on maps not used during shortest-path training to assess spatial transfer (blue bars). For length scaling, we evaluate both the training map (red bars) and unseen maps (orange bars) on start–end pairs whose shortest paths are longer than those observed during training. We observe that language models achieve consistently high success rates (above 90%) on spatial transfer (blue bars), but performance degrades sharply under length scaling, regardless of whether spatial transfer is present (orange bars) or not (red bars).

This length scaling failure is surprising. Intuitively, if a model can reliably solve smaller problems (i.e., shorter paths), one might expect it to decompose a larger problem into small segments to solve. We thus consider two possible explanations:

- **Hardness accumulation.** As shorter paths are not perfectly solved (i.e., ID length success rate $\neq$ 100%), longer paths may fail simply because they are more likely to contain difficult subproblems.
- **Recursive instability.** Even when shorter segments are individually solvable, the model may fail to stably compose them under longer recursive generation.

To disentangle these effects, we decompose long-path success rates, $\Pr(\text{Long})$. For each test path of length exceeding training length, we split it into two subpaths, $\text{Sub}_1$ and $\text{Sub}_2$, each within training length. Then, by the law of total probability,

$$\Pr(\text{Long}) = \Pr(\text{Long} \mid \text{Sub}_1 \wedge \text{Sub}_2) \Pr(\text{Sub}_1 \wedge \text{Sub}_2) + \Pr(\text{Long}, , \neg(\text{Sub}_1 \wedge \text{Sub}_2)).$$

Here, $\Pr(\text{Sub}_1 \wedge \text{Sub}_2)$ is the probability that both subpaths are solved correctly, while $\Pr(\text{Long} \mid \text{Sub}_1 \wedge \text{Sub}_2)$ measures the probability of correctly generating the full path when both subpaths are individually solvable. Results in Table 1 show that the complementary term $\Pr(\text{Long}, \neg(\text{Sub}_1 \wedge \text{Sub}_2))$ is small and relatively stable across length groups, compared to the dominant term $\Pr(\text{Long} \mid \text{Sub}_1 \wedge \text{Sub}_2) \Pr(\text{Sub}_1 \wedge \text{Sub}_2)$. Focusing on the dominant term, we observe that $\Pr(\text{Sub}_1 \wedge \text{Sub}_2) \approx \Pr(\text{Sub})^2$. That is, hardness accumulation arises primarily as a multiplicative probability effect from imperfect subpath performance. Given the high subpath performance, the resulting decrease in $\Pr(\text{Sub}_1 \wedge \text{Sub}_2)$ is modest as length increases ($0.846 \to 0.796$). In contrast, the drop in $\Pr(\text{Long} \mid \text{Sub}_1 \wedge \text{Sub}_2)$ is substantially larger ($0.811 \to 0.589$), showing that ***length scaling failure is dominated by compositional instability***.

> **Takeaway 1:** Language models exhibit systematic spatial generalization on composable SOPs, but this success does not extend to length scaling: even if a model can solve smaller instances, it struggles to stably compose them to solve larger ones.

In the following sections, we analyze how training data, training paradigms, and inference strategies affect the generalization on composable SOPs.

## 4    EFFECTS OF DATA SELECTION ON SPATIAL TRANSFER

We start by analyzing the effects of data selection for the classic SFT paradigm. We ask: *how to allocate a fixed training budget of records to best support transfer?* Should the budget go toward collecting diverse answers for each question, or toward covering as many distinct questions as possible (Section 4.1)? What kinds of questions should be prioritized (Section 4.2)?

### 4.1    MORE QUESTIONS VS. MORE ANSWERS

In many domains of current interest (e.g., mathematics, program synthesis, navigation), a single problem naturally admits multiple valid solutions. This makes budget allocation an important consideration in SFT, especially since collecting high-quality solutions often requires significant efforts (Cobbe et al., 2021; Hendrycks et al., 2021). The question is not trivial: the model requires sufficiently diverse questions to capture the underlying rules; but if each problem is paired with only one solution, the model may overfit to surface patterns rather than acquiring the underlying rule, potentially harming transfer. We therefore investigate whether allocating budget to solution diversity improves spatial transfer generalization, or if prioritizing distinct questions is more effective.

**Experiment Design.**    We consider five training budgets $B \in \{5\%, 10\%, 20\%, 60\%, 80\%\}$ of the total possible training records, where the total is determined by the maximum number of directed start–end pairs within the designated training region (80% of the nodes in the training map $G$, as illustrated in Section 2). We use a $50 \times 40$ sparse grid map with $|V| = 2000$ nodes as $G$. For each budget, we vary the number of distinct questions (unique start–end pairs) and the average number of answers per question (distinct valid shortest paths between each node pair), subject to the constraint $N_{\text{questions}} \times N_{\text{answers per question}} = B.$[2] Transfer capability is measured by the success rate (SR, Equation (2)) on disjoint test maps, restricted to paths within the training length (i.e., excluding length-scaling). We evaluate on three spatially disjoint maps of varying size ($30 \times 30$, $40 \times 40$, $50 \times 50$), sparsity (25%–75%), and adjacency. We report the average SR across them.

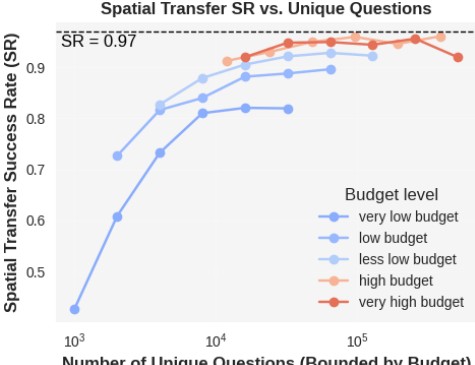

Figure 2: Spatial transfer success rate (SR) improves consistently with more budget allocated to unique questions (log scale). Curves show five budget levels (very low to very high: 5%, 10%, 20%, 60%, 80% of all possible records). Dashed line marks the SR ceil.

**Unique questions drive transfer.** We first confirm that the model can spatial transfer: even when trained on a limited subset of the training map (e.g., 20% of the 80% training region, i.e., 16% of the full training map), it achieves an average success rate of 94% over three spatially disjoint test maps. As shown in Figure 2, under a fixed budget, training on more distinct questions consistently improves transfer. For example, with a low budget, allocating all data to distinct questions with one solution each yields an SR of 94%, compared to only 82% when using fewer questions but 32 solutions per question. This pattern holds across all budget levels, showing that unique questions provide higher marginal value than unique solutions. (This does not imply that solutions are unimportant; rather, one high-quality solution per question appears sufficient under SFT.) However, the benefit of adding more questions quickly saturates: at very high budgets, training with hundreds of thousands of additional questions produces almost no gain over low budgets.

**Takeaway 2:** Spatial transfer is best supported by covering as many distinct questions as possible. This makes the most of the training budget, especially when collecting solutions is expensive.

---

[2]If a question admits fewer distinct solutions than required, we include all available solutions and allocate the remaining budget to other questions without repetition.

## 4.2 COVERAGE VS. DIVERSITY IN QUESTIONS

If distinct questions matter more than multiple solutions, **which kinds of questions should be prioritized?** Prior work on generalization has long emphasized two training distribution properties, *coverage* and *diversity*. (Lake & Baroni, 2018; Bahdanau et al., 2018; Keysers et al., 2019; Lippl & Stachenfeld; Levy et al., 2023; Ahuja & Mansouri, 2024). While commonly believed to matter, their precise role remains unclear: **are higher coverage and diversity always beneficial? How do they interact?** We next vary these two factors to examine their effect on systematic transfer. Following Chang et al. (2025), we define them in questions over node primitives:

**(Local) Coverage.** Coverage measures the fraction of unique nodes (i.e., primitives) in the *local training map* $G = (V, A)$ that appear in the training set. Formally, following Section 2, let $V_{\text{train}} \subseteq V$ denote the set of nodes included in $\mathcal{D}_{\text{train}}$. We define $c = |V_{\text{train}}|/|V|$, which ranges between 0 and 1.

*Remark.* We stress that coverage is defined **only locally relative to the training map, not the global universe.** Even $c = 0.8$ corresponds only to a tiny fraction of the universe of possible primitives. Since the model is expected (and observed) to spatially transfer to (infinitely) many disjoint maps $\hat{G} = (\hat{V}, \hat{A})$, including nodes from all such maps in the denominator would only dilute the fraction and make coverage misleadingly small. For comparability, we therefore compute coverage solely with respect to the training map; any nodes in any disjoint map $\hat{G}$ are in fact uncovered.

**Diversity.** Diversity measures how richly the observed nodes are combined in training. Formally, recall from Section 2 that $\text{supp}(\mathcal{D}_{\text{train}})$ denotes the set of ordered node pairs included in training. We define $d = |\text{supp}(\mathcal{D}_{\text{train}})|/|V_{\text{train}}|$, which ranges from 1 to $|V| - 1$. Intuitively, $d$ corresponds to the average number of distinct endpoints $j$ that each start node $i \in V_{\text{train}}$ is paired with. In practice, we control diversity explicitly by constraining, for each $i$, the number of distinct $j$'s that appear in pairs $(i, j)$.

*Remark.* Note that we intentionally do not normalize $d$ by $|V_{\text{train}}| - 1$, since $|V_{\text{train}}| = c|V|$; this would couple diversity with coverage and prevent the two from being varied independently.

**Experiment Design.** To disentangle the roles of coverage and diversity, we design controlled experiments where one factor is varied while the other is fixed. Coverage is defined as $c = |V_{\text{train}}|/|V|$ and is varied by *linearly* increasing the fraction of nodes included in the training questions from as low as 4% up to 80% of the nodes in the training map. Diversity $d$ is varied by controlling how many distinct endpoints $j$ each start node $i$ is connected to, ranging *exponentially* from $2^0$ to $2^7$. We control the total number of question–answer records to remain fixed across conditions. We use the same evaluation protocol as before, with one training map $G$ and three independent and disjoint maps $\hat{G}$, and report the average performance (measured by the success rate) over the three test maps.

**Main Results.** We summarize three selected observations from Figure 3. Marginal trends of each factor (see Figures 10 and 11) and their individual analysis are deferred to Section D.5.

**(1) Coverage sets the ceiling of spatial transfer.** Across a wide range of diversity levels (from $d = 2^2$ to $2^7$), the curves converge to a similar maximum SR once coverage is sufficiently high (e.g., $\approx 0.90$ for $c \geq 64\%$). When coverage is low, even exponentially increasing diversity fails to achieve strong transfer. In contrast, high coverage can partially compensate for low diversity.

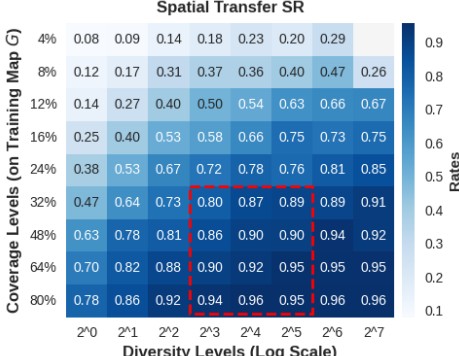

Spatial Transfer SR

Figure 3: Interaction between coverage and diversity on problem-solving transfer.

**(2) Minimal diversity is required, but excessive diversity can hurt at low coverage.** At very low diversity ($d = 1, 2$), SR grows slowly as coverage increases and saturates at a noticeably lower level. Only when diversity passes a small threshold (e.g., $d \geq 2^2$) does coverage begin to unlock its full effect. However, excessively high diversity can sometimes reduce success rates when coverage is low (e.g., $c \leq 8\%$).

**(3) A resource-efficient regime** (highlighted in red, Figure 12) is to target mid-to-high coverage ($\geq 32\%$) with modest diversity (8–32) because diversity grows in cost exponentially.

Table 2: Performance of different data regimes across MathQA categories.

|  | | probability (*easy*) | gain (*medium*) | physics (*hard*) |
|---|---|---|---|---|
| **Qwen2.5-7B-Instruct** | – | 0.729 | 0.70 | 0.68 |
| **More questions** | High Coverage | **0.792** | **0.82** | **0.77** |
|  | High Diversity | 0.792 | 0.74 | 0.74 |
| **More solutions** | – | 0.771 | 0.72 | 0.70 |

Due to space constraints, we present only the core findings in the main text. For a complete and nuanced analysis of coverage–diversity effects, including marginal and interaction results, readers are strongly encouraged to refer to Section D.5.

> **Takeaways 3–5 (Brief).** Coverage determines the ceiling of spatial transfer. A minimal level of diversity is required to approach this ceiling, but beyond a small threshold, diversity yields diminishing returns and may even hurt when coverage is low. Insufficient coverage cannot be compensated by increasing diversity. Accordingly, the most cost-efficient regime combines mid-to-high coverage with modest diversity.

**Mechanistic explanation for the success of spatial transfer.** We observed strong generalization to disjoint maps, akin to how a language model, once having internalized a rule in English, can seamlessly apply the same rule to other languages it already knows. This suggests that the model does not merely memorize surface-level node n-gram, but rather encodes structured latent operators that can be flexibly reused across domains—for instance, "move to an adjacent node towards the end node" heuristic (which we probed in Section D.2). This interpretation aligns with recent theoretical progress framing attention as a hypernetwork (Schug et al., 2024), where attention scores serve as latent codes parameterizing reusable computations.

**Practical insights.** In real-world problem-solving, training can be viewed as teaching the model to construct directed acyclic graphs (DAGs) over question-type-specific sets of elements, with rules defined on each set (conceptual skills). The element set can therefore be treated as the primitive shared across questions. Under this view, *coverage measures how many distinct element sets appear in training*, while *diversity measures how many distinct directed structures are supported per element set*. Takeaways 2-5 provide concrete guidance for dataset selection: to enable systematic transfer under limited budgets, one should prioritize broad coverage of element sets in questions, combine it with only modest diversity in their structural realizations, and spend the least effort on solution diversity. A concrete example in math is provided below.

## 4.3    A CASE STUDY IN THE MATH DOMAIN

To examine whether the conclusions drawn from our controlled navigation environment generalize to a more realistic setting, we conduct a case study on mathematical reasoning using the **MathQA** dataset (Amini et al., 2019). Each MathQA problem is annotated with a *linearized operation program*, which allows us to extract the element sets by converting each program into an *unordered multiset of operations* (i.e., operation set). Different programs may share the same element set. See Section E for exmaples. We then define coverage and diversity over these operation sets.

**Setup.**    We fine-tune Qwen2.5-7B-Instruct (Team, 2024) on three representative categories—`probability` (easy), `gain` (medium), and `physics` (hard)—under a strict data budget of roughly 1,000 samples per category (and only $\sim 200$ for `probability` due to its very small size). We use DeepSeek-R1 (DeepSeek-AI et al., 2025) to generate high-quality chain-of-thought solutions for supervision. Following our earlier observations, we compare the following data allocation strategies (more details are provided in Section E):

- **More Questions:** one solution per question, maximizing the number of distinct questions. This strategy has two variants:
    - **High Coverage**: maximize the number of distinct operation sets;
    - **High Diversity**: increase the number of questions per operation set (tenfold), and therefore operate under a smaller coverage.
- **More Solutions:** ten solutions per question and reducing the number of distinct questions.

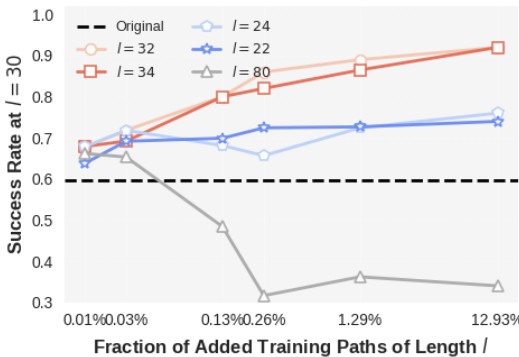 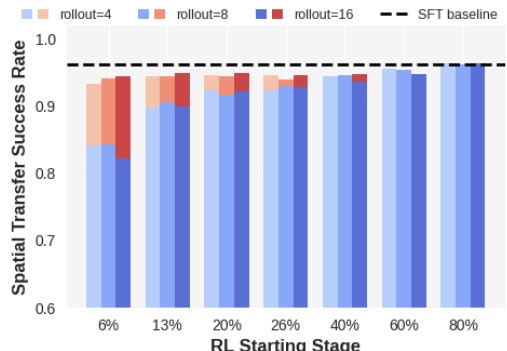

Figure 4: Effect of adding paths of different lengths on SR to length $= 30$. A few neighboring-and-longer paths (e.g., $l = 32, 34$) rescue performance, shorter ones ($l = 22, 24$) give little gain, and overly long ones ($l = 80$) degrade it. Dashed line: no augmentation.

Figure 5: RL does not further improve spatial transfer: performance bounded by the SFT baseline. Each group of bars corresponds to a different SFT checkpoint used to initialize RL. Blue bars denote one-pass RL and red bars denote multi-pass RL.

**More questions consistently outperform more solutions.** Across all three categories, both *More Questions* regimes (High Coverage and High Diversity) achieve better generalization than *More Solutions*. Notably, these improvements appear under an extremely small training budget: roughly 1,000 samples for category `gain` and `physics`, and only $\sim 200$ for `probability`). Despite such limited supervision, allocating more budget to distinct questions still produces clear performance gains. For instance, in the `gain` category, accuracy rises from $0.70$ to $0.82$ under `High Coverage`, and a similar increase appears in the harder `physics` category ($0.68 \rightarrow 0.77$).

**Coverage plays the dominant role.** Within the *More Questions* group, *High Coverage* consistently outperforms *High Diversity* (e.g., $0.82$ vs. $0.74$ in `gain`; $0.77$ vs. $0.74$ in `physics`). This suggests that encountering a broader range of conceptual skills matters more than exposing the model to many different ways of applying or combining the seen skills. Taken together, these findings reinforce a simple intuition: ***under realistic data budgets, breadth matters more than depth***.

## 5 EFFECTS OF DATA SELECTION ON PROBLEM-SOLVING SCALING

In addition to problem-solving transfer, a fundamental dimension of extrapolation is *problem-solving scaling* (Sinha et al., 2024; Hupkes et al., 2020; Cai et al., 2025). As shown in Figure 1, even the strongest spatial-transfer model exhibits severe length scaling failure[3]. While generalization within the training length regime is nearly perfect, performance rapidly deteriorates once path length exceeds the training maximum. This raises a key question: **if the data conditions that enable spatial transfer** (e.g., sufficient training questions and high primitive coverage) **do not suffice for scaling under SFT, what additional data conditions are required?**

**Rescuing with slightly longer paths.** Surprisingly, we found that adding even a handful of training examples randomly sampled from lengths at or slightly above the target length can substantially rescue performance, whereas adding shorter paths provides much less benefit. For instance, in Figure 4, we evaluate performance on target length $= 30$, where the model exhibits suboptimal generalization. Augmenting the training set with a very small fraction ($\approx 1\%$ of the training data) of slightly longer paths (e.g., $l = 32, 34$) raises success rates to nearly 90%. By contrast, adding shorter paths (e.g., $l = 22, 24$) yields small gains—even when added in large amounts (12%)—while much longer paths (e.g., $l = 80$) confuse the model and degrade performance. These results suggest that, under SFT, curriculum-like exposure to slightly longer examples provides the critical adaptation signal that neither shorter nor excessively long paths can supply.

> **Takeaway 6:** Length generalization can be rescued by adding *slightly longer* paths; shorter ones give little benefit, and overly long ones can even harm performance.

---

[3]We test scaling on node pairs requiring longer paths than those seen during training (details in Section D.6).

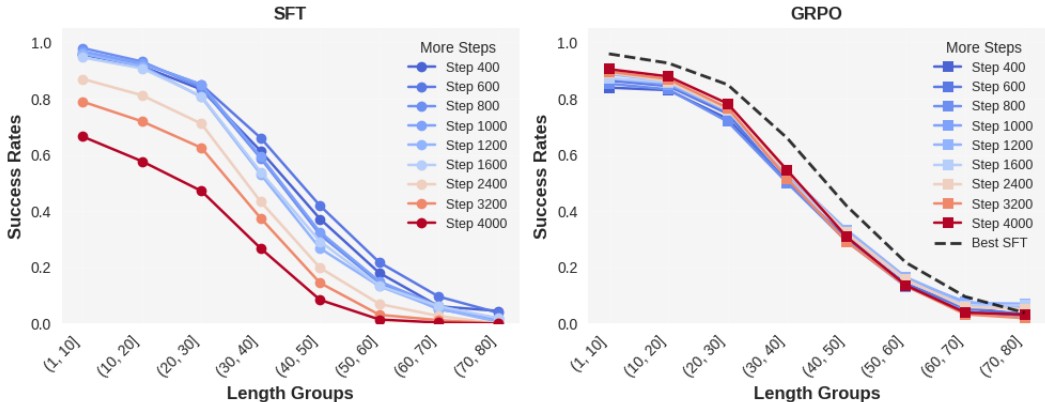

Figure 6: Length scaling under extended training (1 epoch ≈ 400 steps). Left: SFT improves at first but quickly overfits with more epochs. Right: RL (GRPO) remains stable across epochs but never exceeds the best SFT bound (dashed line).

## 6  EFFECTS OF TRAINING PARADIGM ON PROBLEM-SOLVING

While the previous sections focus on how data properties shape problem-solving skills, another natural question is whether the training paradigm itself can provide further gains. A recent line of work presents compelling empirical evidence that reinforcement learning (RL) can enable extrapolative generalization beyond supervised fine-tuning (SFT) (Chu et al., 2025; Chen et al., 2025; Huang et al., 2025). At the same time, other studies argue that RL primarily unlocks capabilities already present in SFT rather than introducing new ones (Yue et al., 2025a; Ma et al., 2025). We therefore test whether RL adds value on top of SFT for spatial transfer and length scaling.

***Spatial transfer setup.*** As detailed in Section 2, we train the RLVR model using an unbiased GRPO variant with a binary reward of 1 if the generated sequence forms a valid shortest path, and 0 otherwise. The training data budget is set to high, under which the model is capable of spatially transferring (see Figure 2). RL is warm-started from different SFT checkpoints, ranging from 6% to 80% of SFT training progress. For each warm-start, we vary the number of rollouts per prompt in $4, 8, 16$. We report two types of RL outcomes: (i) **one-pass RL** (blue bars), where the model is trained on the remaining data for a single pass; and (ii) **multi-pass RL** (red bars), where RL is allowed to repeatedly reuse the same remaining data. This disentangles the effects of warm-start quality, data availability, and rollout compute.

**RL does not improve spatial transfer.** As shown in Figure 5, RL does not confer additional capabilities beyond what can be achieved by a fully trained SFT for spatial transfer: the best RL curves are always bounded by the SFT upper line. Early warm-starts perform poorly in one-pass RL (blue bars), but multi-pass training (red bars) can recover the gap

***Length scaling setup.*** To test whether RL can address the known weakness of SFT in length scaling under "unlimited" passes, we continue RL training for up to ∼20 epochs on the same dataset (rollout fixed at 8), with the model warm-started from an early SFT checkpoint (after 1 epoch, 400 steps). For comparison, we also extend SFT training for the same number of epochs.

**RL stabilizes training but cannot exceed the best SFT.** Figure 6 compares SFT and RL under the same 10-epoch progress and show a clear pattern: SFT initially improves with more steps but quickly overfits, leading to sharp degradation. RL curves, in contrast, remain tightly clustered across steps, indicating stable training even after many epochs. However, RL never exceeds the best SFT bound, confirming that additional training (whether SFT or RL) cannot resolve the fundamental limitation in length scaling. Results for extended RL training up to 20 epochs are provided Section D.7 and show the same stable trend.

Across both settings, RL primarily serves to ***stabilize training*** and mitigate overfitting during prolonged optimization, rather than unlock new reasoning capabilities. Error-type analysis in Section G further supports this: *SFT and RL exhibit nearly identical error types and distributions*, indicating that RL cannot correct errors inherent to the corresponding SFT model. Consequently, the performance ceiling is determined by the best SFT model. This behavior is consistent with recent analyses of ***generation–verification gap (Swamy et al., 2025)***: RL provides benefits when generating good

continuations is difficult but verifying them is easy. In our setting, the optimal path can be computed explicitly, making generation nearly as easy as verification and effectively closing this gap. When data are sufficient and well-designed, SFT achieves higher efficiency, while RL functions as a robust fallback that trades peak performance for stability.

> **Takeaway 7:** RL stabilizes training and prevents overfitting but does not unlock new transfer or scaling capabilities. The performance ceiling is always set by the best SFT model. SFT is more efficient with sufficient, high-quality data, while RL provides a safer default when principled data selection is missing.

*Remark.* Our findings do not suggest that RL is useless. In practice, training data are often noisy, heterogeneous, or subject to domain shifts, where SFT may overfit or struggle. In such settings, RL can generalize better by optimizing sequence-level rewards—not because it extends the capability frontier, but because it maintains robustness where SFT degrades.

## 7 EFFECTS OF INFERENCE-TIME STRATEGIES ON PROBLEM-SOLVING

Recent work shows that reasoning performance can be substantially improved at inference time by allocating additional compute (test-time scaling), typically by generating and selecting among multiple reasoning trajectories, such as Self-Consistency (Wang et al., 2022), best-of-$N$ sampling (Brown et al., 2024), and structured search like Tree-of-Thought (Yao et al., 2023). This raises the question of **whether observed length-scaling failures stem from insufficient search (i.e., a failure to surface latent capabilities encoded in the model) rather than intrinsic model limitations**.

To rule out this possibility, we consider two strategies in addition to *greedy decoding*: (1) *majority-of-10*: sample 10 trajectories and select the most frequent output (corresponding to a standard Self-Consistency procedure (Wang et al., 2022)); (2) *shortest-of-10*: sample 10 trajectories and select the shortest one (an objective-guided selection strategy that explicitly exploits knowledge of the task reward). For each length group, we evaluate both SFT and RL (GRPO) models under all three inference settings. All stochastic strategies use 10 independent samples with identical temperature.

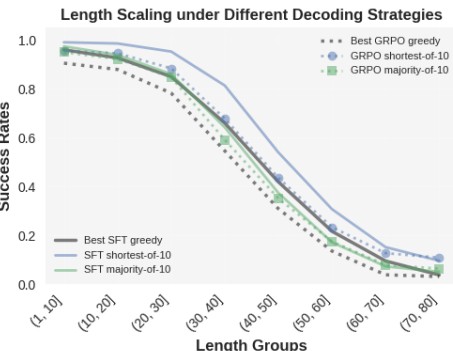

Figure 7: Length scaling under different inference-time strategies. Inference-time search improves SR but does not rescue scaling failures; RL remains below SFT and appear to contract the effective solution space.

**Results.** As shown in Figure 7, stronger inference-time strategies (e.g., Shortest-of-10) can improve empirical success rates for both SFT and RL models. However, the overall degradation trend remains largely unchanged; enhanced test-time search shifts the performance curve upward but does not fundamentally alter the length scaling failure.

Moreover, RL models consistently underperform their SFT counterparts under each inference setting. Notably, even with the strongest objective-guided sampling (Shortest-of-10), RL models achieve performance comparable to SFT greedy decoding and remain substantially below SFT Shortest-of-10. This suggests that RL training may restrict the effective solution space, limiting the diversity of trajectories available for inference-time selection.

## 8 CONCLUSIONS

Our work studies systematic generalization in composable sequential optimization problems within a controlled synthetic pathfinding environment. We reveal a clear asymmetry: models transfer structurally across unseen maps but fail under length scaling due to recursive instability. By disentangling training data, training paradigms, and inference-time strategies, we systematically analyze the contribution of each factor to generalization performance. Our findings offer a unified view of how different stages of the learning pipeline shape generalization in language models.

ACKNOWLEDGMENTS

Jiayuan Ye is supported by the Apple Scholars in AI/ML PhD fellowship.

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

## Appendix Contents

## A    LIMITATIONS

The primary limitation of our study is that most results are derived from a controlled synthetic testbed with relatively small models, which may raise concerns about practical relevance. This reflects an inherent trade-off: narrowing the problem scope enables rigorous and well-controlled analysis, but inevitably reduces realism. While large-scale practical benchmarks are abundant, they often make it difficult to isolate the source of generalization behavior. Our approach therefore prioritizes control and interpretability over scale.

To partially address this limitation, we validate key findings on training data properties using math datasets on 7B-scale models and relate our RL observations to recent theoretical developments. Moreover, sequential optimization itself constitutes a broad class of real-world tasks, and path-based data is closely connected to reasoning and mathematical problem solving.

## B    LLM USAGE

We use ChatGPT and Gemini to support writing and formatting, such as grammar and style refinement, polishing figure and table captions, and other surface-level edits. Some of the code is written with the help of GitHub Copilot and Claude, for example, in code auto-completion and providing debugging suggestions.

## C    RELATED WORKS

**Compositional and length generalization**    Our notion of extrapolative problem-solving is closely tied to systematicity in compositional generalization (CG) and to length generalization, sometimes referred to as productivity in CG. Compositional Generalization (CG) plays a central role in generalization studies, which underpins the ability to extend learning to unseen situations (Hupkes et al., 2020). *Systematicity*, the most common definition of CG, refers to the capacity to systematically recombine known primitives and rules (Dankers et al., 2021). While Systematicity has long been regarded as a fundamental challenge for neural networks (Liška et al., 2018; Lake & Baroni, 2018; Loula et al., 2018; Csordás et al., 2022; Ontanon et al., 2021; Keysers et al., 2019; Lewis et al., 2022), recent work has increasingly provided evidence that modern generative models exhibit nontrivial Systematic CG abilities (Lepori et al., 2023; Yun et al., 2022; Okawa et al., 2023; Ramesh et al., 2023; Abedsoltan et al., 2025; Xu et al., 2024). Further progress in understanding Systematic CG comes from multiple perspectives: the structural side (Lepori et al., 2023; Schug et al., 2024; Quirke & Barez, 2023; Li et al., 2023a), the task side (Abedsoltan et al., 2025; Zhou et al., 2023), and the data side (Lippl & Stachenfeld; Ahuja & Mansouri, 2024; Kamb & Ganguli, 2024; Chang et al., 2025; Cagnetta et al., 2024). For example, (Schug et al., 2024) shows that multi-head attention can function as a hypernetwork supporting compositional behavior (e.g., encouraging learning functions as reusable components). From the data perspective, (Ahuja & Mansouri, 2024) derives provable guarantees for length and compositional generalization under sufficient training-set diversity, while (Chang et al., 2025) frames training data coverage as a key factor in a model's ability to generalize to unseen combinations.

Progress on compositionality in vision object learning has become increasingly well characterized, both empirically (Yun et al., 2022) and theoretically (Wiedemer et al., 2023b;a). In language, however, understanding remains fragmented: studies have pointed to a variety of factors (e.g., from model-side (Kazemnejad et al., 2023; Petty et al., 2023) to data-side (Ahuja & Mansouri, 2024; Chang et al., 2025)), but lacking an integrated account. Our work takes a data-centric perspective, unifying the recurring factors into a coherent view of how they jointly shape the model's systematic extrapolation. Inspired by progress in vision, where disentangled primitives and rules have enabled clearer advances, and to avoid prior inconclusive results in language (Lake & Baroni, 2018; Furrer et al., 2020; Dziri et al., 2023), we design map-navigation tasks in which primitives (nodes) and rules (mobilities) are cleanly disentangled, allowing us to directly assess the influence of data properties on generalization performance (Liang et al., 2025).

*Length generalization*, or *Productivity*, is another notion within the broader study of compositional generalization (Sinha et al., 2024). It has been widely discussed as a central challenge (Dubois et al., 2019; Newman et al., 2020; Cai et al., 2025; Fan et al., 2024; Jelassi et al., 2023; Anil et al.,

2022), and is sometimes framed as a form of recursive composition or extrapolation (Kim & Linzen, 2020; Hupkes et al., 2020; Dziri et al., 2023). For instance, in natural language tasks, longer input sequences may correspond to recursive or nested structures of previously seen phrases (Kim & Linzen, 2020). In our setting, path length provides a directly controllable axis for studying this phenomenon: extrapolating to longer paths mirrors the core difficulty of length generalization, while allowing us to systematically manipulate the data properties and training paradigm to probe its limits.

**Graph navigation and other capabilities**    While our work may appear related to prior studies that evaluate models' graph navigation abilities (Zhang et al., 2024; Wang et al., 2025a), build powerful graph models (Wang et al., 2025b; Yehudai et al., 2021), or use graph data to enhance LMs' reasoning (Zhang et al., 2025), it is in fact fundamentally different in both task setting and goal. First, rather than treating the graph as the task itself (i.e., providing the model with many small graphs in prompts and training it to solve specific navigation task on future graphs), our work considers the large map and treats each map as an independent vocabulary world. Instead of explicitly describing the graph structure, we require the model to learn the connections and the map itself, analogous to how LLMs acquire word semantics during pretraining. The map is sufficiently complex that it cannot be memorized or learned within a single prompt. Second, our goal is not to test whether models can perform navigation tasks, nor to improve navigation performance by modifying architectures or training pipelines. Instead, we seek to understand models' compositionality/extrapolation under varying data distributional properties. To ensure that our focus remains on distributional effects, we even restrict ourselves to tasks that are already proven to be learnable (Cohen et al., 2025; Dai et al., 2024). Therefore, our work is also orthogonal to studies that examine whether models can perform specific capabilities with certain heuristics under narrowly defined tasks (Quirke et al., 2024; Nikankin et al., 2024; Cohen et al., 2025).

## D    ADDITIONAL RESULTS

### D.1    IMPLEMENTATION AND LICENSING.

Our LLaMA-style models are based on the standard implementations in the Hugging Face `transformers` library (Apache 2.0 license) (Wolf et al., 2020). Reinforcement learning with Dr.GRPO is conducted using the `GRPOTrainer` from the Hugging Face TRL library (Apache 2.0 license) (von Werra et al., 2020).

**Pretrain Specifications.**    We pretrain the model on random-walk trajectories to provide basic "map semantics" without leaking any shortest-path information. The pretraining data consists of long random walks sampled uniformly across the grid.

We adopt a pretraining corpus of **10M random-walk trajectories** (approximately 1.3B tokens), trained for **124,999** steps. This number was chosen based on preliminary runs with smaller datasets (2M, 5M, and 8M trajectories), where we observed that the model's valid-path rate increases steadily with data size and saturates at the 10M scale. As reported in Table 4, at this final budget, the pretrained model achieves a **valid-path rate of** 1.0 while retaining zero shortest-path capability, confirming that pretraining captures structural map knowledge without imparting any optimal navigation behavior.

### D.2    PROBING: MODEL TRACKS DISTANCE TO THE END NODE

We investigate whether the model encodes the remaining shortest-path distance to the end node, which would allow it to apply heuristics such as "move towards the goal". For probing, we apply a 2-layer MLP, $p_\theta(x_t^k) = \text{softmax}\big(W_1 \text{ReLU}(W_2 x_t^k)\big)$, where $x_t^k$ denotes the hidden representation of the $t$-th token at the $k$-th layer. The probe outputs a probability distribution over discretized distance classes ($C = 10$). Although we probe at a *fixed token position*, the hidden state at this position already integrates information from all previous tokens, including the traversed path. We thus train a probe on paths of varied length from the training map for each layer, and test it on paths from a disjoint map, grouping path lengths from 1–20 into 10 classes (granularity of 2). As shown in Table 3, the nonlinear probe achieves high accuracy, especially in middle-to-late layers, supporting the hypothesis that the model encodes distance-based heuristics as reusable operators for spatial

transfer. While a linear probe would provide a stronger conclusion, we have not yet identified one that performs well in this setting.

Table 3: Probe accuracy (%) across layers.

| Layer | Accuracy (%) |
|-------|--------------|
| 0 | 35.94 |
| 1 | 32.78 |
| 2 | 57.85 |
| 3 | 76.58 |
| 4 | 83.14 |
| 5 | **86.29** |
| 6 | **85.77** |
| 7 | 81.55 |

### D.3 PRETRAINING DOES NOT INTERFERE WITH DOWNSTREAM SHORTEST-PATH LEARNING

To ensure that our pretraining stage does not leak or overlap with the downstream shortest-path task, we evaluate pretrained models directly on shortest-path generation. Both the loss distribution analysis Figure 8 and generation performance Table 4 confirm that pretraining does not endow the model with shortest-path capabilities, thereby ruling out interference.

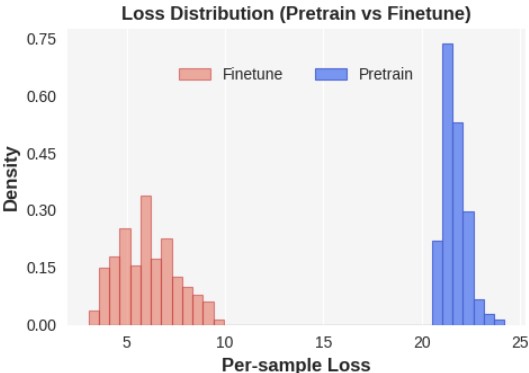

Figure 8: Loss distributions of the pretrained and fine-tuned models on test (*i.e.* unseen) shortest paths. The distributions are completely disjoint, indicating that pretraining alone does not prepare the model with shortest-path generation capabilities.

### D.4 TRAINING PATHS LENGTH DISTRIBUTION UNDER VARYING COVERAGE VALUES

To examine whether the shortest-path distance distribution shifts as coverage increases, potentially contributing to performance gains, we plot the shortest-path length histograms for different coverage ratios (under fixed diversity). Figure 9 reports the relative-frequency distributions ($x$: path length, $y$: proportion of samples) for coverage values $\{0.01, 0.05, 0.1, 0.2, 0.6, 0.8\}$ on the training map $G$.

Table 4: Performance on shortest-path generation. Pretrained models cannot generate valid shortest paths, confirming that pretraining does not interfere with downstream learning. The Avg. Length Ratio measures the ratio between the true shortest-path length and the generated path length.

| Model Trained on | Valid Path Rate↑ | Shortest Path Rate↑ | Avg. Length Ratio↑ |
|------------------|------------------|---------------------|--------------------|
| Pretrain | 1.0 | 0.00 | 0.0707 |
| Finetune | 1.0 | 0.9726 | 0.9983 |

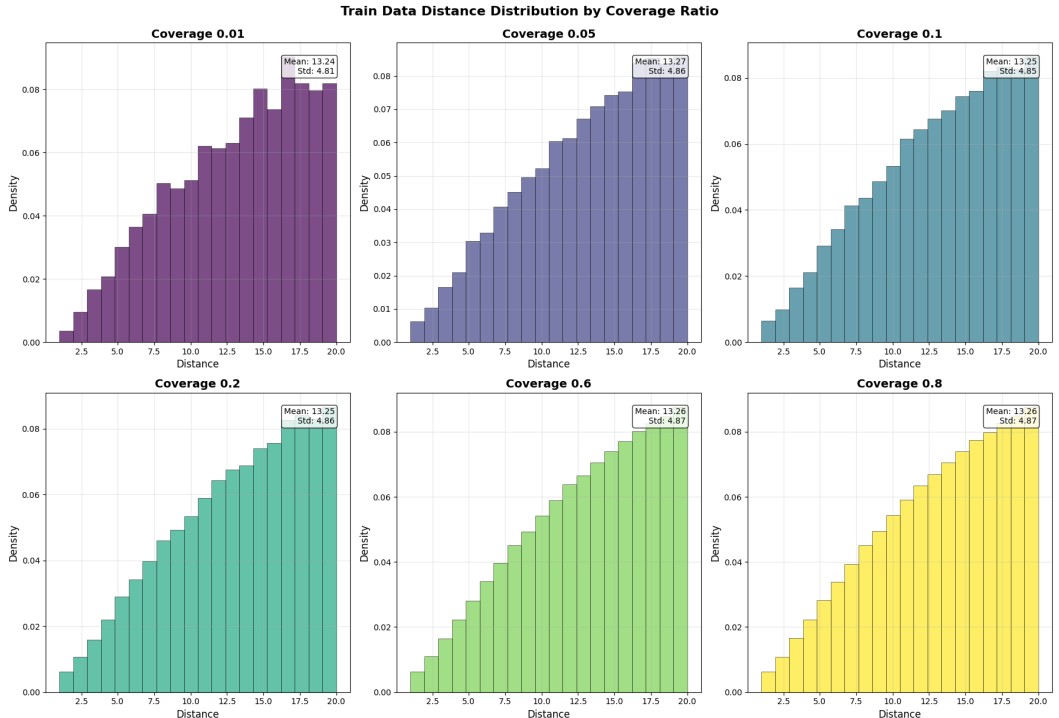

Figure 9: Shortest-path lengths distribution under varying coverage ratios (fixed diversity). The distributions remain stable across settings, indicating that coverage does not alter distance exposure.

Although the total number of sampled start–end pairs increases with coverage, the **shape of the path-length distribution remains highly stable** across all settings. The mean shortest-path length is consistently around $13.25$ with a standard deviation of approximately $4.85$, and the proportion of samples near the maximum observed training length ($L_{\max} = 20$) shows minimal variation.

These statistics confirm that increasing coverage does not introduce systematic changes to the length distribution, ensuring that our analyses isolate the effect of coverage itself rather than incidental differences in distance exposure.

### D.5 EXTENDED ANALYSIS OF COVERAGE AND DIVERSITY

If more distinct questions are more valuable, **which kinds of questions should be prioritized?** Prior work on generalization has long emphasized the importance of training distribution properties such as *coverage* and *diversity*. These factors have been discussed since early seq2seq RNN and CNN models (Lake & Baroni, 2018; Bahdanau et al., 2018; Keysers et al., 2019), and continue to play a central role for decoder-only Transformers (Lippl & Stachenfeld; Ahuja & Mansouri, 2024; Levy et al., 2023). However, while commonly believed to matter, their precise role remains unclear: **Are higher coverage and diversity always beneficial? How do they interact?** In this section, we empirically vary these two classic factors in a decoupled way to measure their effect on systematic transfer. We begin by defining these notions in our setting. Following Chang et al. (2025), we define coverage and diversity in questions over node primitives.

**(Local) Coverage.** Coverage measures the fraction of unique nodes (i.e., primitives) in the *local training map* $G = (V, A)$ that appear in the training set. Formally, following Section 2, let $V_{\text{train}} \subseteq V$ denote the set of nodes included in $\mathcal{D}_{\text{train}}$. We define $\text{c} = |V_{\text{train}}|/|V|$, which ranges between 0 and 1.

*Remark.* We stress that coverage is defined **only locally relative to the training map, not the global universe.** Even $\text{c} = 0.8$ corresponds only to a tiny fraction of the universe of possible primitives. Since the model is expected (and observed) to spatially transfer to (infinitely) many disjoint maps $\hat{G} = (\hat{V}, \hat{A})$, including nodes from all such maps in the denominator would only dilute the fraction and make coverage misleadingly small.

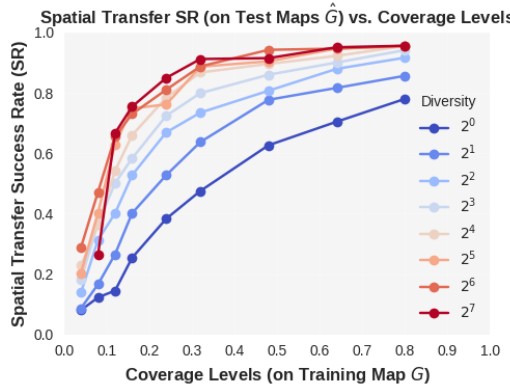 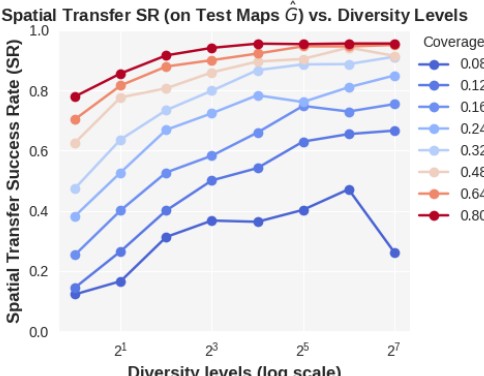

Figure 10: Spatial transfer success rate (SR) measured on the disjoint maps as the nodes coverage ratio in question in the training map increases. Each curve corresponds to a fixed diversity level.

Figure 11: Spatial transfer success rate (SR) measured on the disjoint maps as the node pair composition diversity level increases. Each curve corresponds to a fixed coverage ratio.

For comparability, we therefore compute coverage solely with respect to the training map; any nodes in any disjoint map $\hat{G}$ are in fact uncovered.

**Diversity.** Diversity measures how richly the observed nodes are combined in training. Formally, recall from Section 2 that $\text{supp}(\mathcal{D}_{\text{train}})$ denotes the set of ordered node pairs included in training. We define $d = |\text{supp}(\mathcal{D}_{\text{train}})|/|V_{\text{train}}|$, which ranges from 1 to $|V| - 1$. Intuitively, $d$ corresponds to the average number of distinct endpoints $j$ that each start node $i \in V_{\text{train}}$ is paired with. In practice, we control diversity explicitly by constraining, for each $i$, the number of distinct $j$'s that appear in pairs $(i, j)$.

*Remark.* Note that we intentionally do not normalize $d$ by $|V_{\text{train}}| - 1$, since $|V_{\text{train}}| = c|V|$; this would couple diversity with coverage and prevent the two from being varied independently.

**Experiment Design.** To disentangle the roles of coverage and diversity, we design controlled experiments where one factor is varied while the other is fixed. Coverage is defined as $c = |V_{\text{train}}|/|V|$ and is varied by *linearly* increasing the fraction of nodes included in the training questions from as low as 4% up to 80% of the nodes in the training map. Diversity $d$ is varied by controlling how many distinct endpoints $j$ each start node $i$ is connected to, ranging *exponentially* from $2^0$ to $2^7$. We control the total number of question–answer records to remain fixed across conditions. We use the same evaluation protocol as before, with one training map $G$ and three independent and disjoint maps $\hat{G}$, and report the average performance (measured by the success rate) over the three test maps.

### D.5.1 MARGINAL EFFECTS

We draw two key observations from Figure 10:

**(1) Coverage determines the ceiling of spatial transfer.** Across a wide range of diversity levels (from $d = 2^2$ to $s^7$), the curves converge to a similar maximum SR once coverage is high. This shows that coverage ultimately sets the upper bound of systematic generalization, while diversity only influences how quickly, as coverage increases, this ceiling is approached.

**(2) Minimal diversity is required to unlock efficient use of coverage.** However, at very low diversity ($d = 1, 2$), SR grows slowly and saturates at a noticeably lower level. Only when diversity passes a small threshold ($d \geq 4$ here) does coverage begin to unlock its full effect.

**(3) Threshold and plateau phenomenon.** With sufficient diversity ($d \geq 4$), SR exhibits a sharp inflection around mid-to-low coverage ($\approx 0.2$–$0.25$). Beyond this point, additional coverage yields diminishing returns, whereas below it, generalization remains poor.

> **Takeaway 3:** Coverage in question sets the ceiling of spatial transfer, but minimal diversity is required to unlock it efficiently. Coverage also creates a sharp inflection point in SR, indicating a cost-efficient region at low values.

We next vary diversity while keeping coverage constant. Results in Figure 11 show two main patterns:

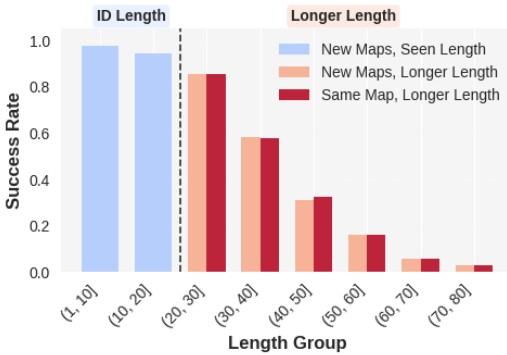

Figure 13: Spatial transfer success and length scaling failure.

**(1) Log-linear gains at mid-to-high coverage.** At mid-to-high coverage, performance grows roughly linearly in $\log(d)$, indicating that the marginal benefit of additional diversity decreases as $d$ grows. In other words, exposing the model to a few diverse combinations is highly beneficial, but each further doubling of diversity yields progressively smaller gains.

**(2) High diversity can hurt when coverage is low.** At low coverage, adding diversity sometimes reduces success rates. This likely occurs because exhaustively combining a tiny set of primitives encourages memorization rather than rule abstraction. For example, if a model is trained on a very small set $1, 2, 3$ and exposed to all possible addition combinations, it can simply memorize the resulting facts without grasping the general rule of addition.

> **Takeaway 4:** Diversity can bring rapid early gains but quickly flattens out, and may even harm transfer when coverage is low.

### D.5.2   COVERAGE–DIVERSITY INTERACTION

Jointly analyzing coverage and diversity (Figure 12) reveals a clear interaction. At low coverage, even exponentially high diversity cannot rescue the performance (e.g., SR rises only from 0.08 to 0.29 when coverage is 4%). By contrast, at higher coverage (e.g., 32%), diversity strongly amplifies performance (raising SR from 0.47 at low diversity to 0.91 at high diversity). High coverage can partially compensate for low diversity (e.g., increasing SR from 0.08 to 0.78 when $d = 1$).

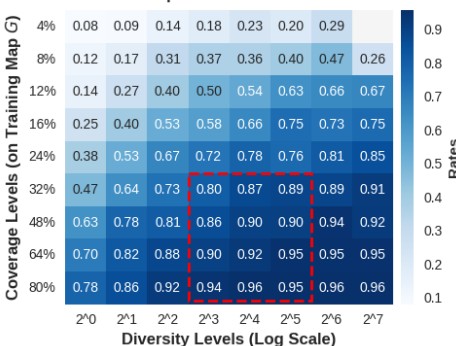

Figure 12: Interaction between coverage and diversity on problem-solving transfer.

Because diversity grows in cost exponentially, **a resource-efficient regime (highlighted in red, Figure 12) is to target mid-to-high coverage ($\geq 32\%$)** with modest diversity (8–32). This achieves strong performance at a much lower computational cost than maximizing both dimensions. Beyond this regime, both coverage and diversity show diminishing returns. Note that dataset size is approximately controlled across conditions by varying the number of answers, which actually makes high-coverage-low-diversity settings relatively disadvantaged (fewer distinct questions). That such settings still outperform low-coverage-high-diversity settings strengthens the conclusion.

> **Takeaway 5:** Low coverage in question cannot be rescued even with extremely high diversity, but low diversity can be compensated by high coverage. Moderate–to-high coverage with modest diversity achieves the best efficiency–performance trade-off.

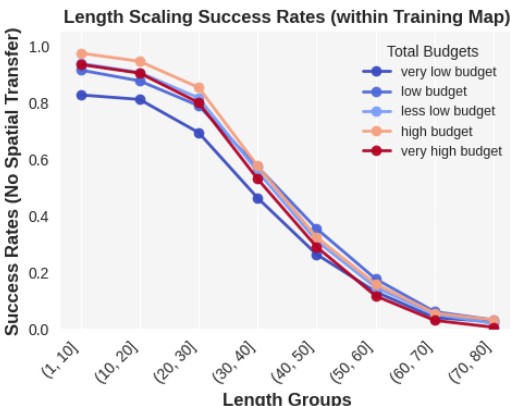

Figure 14: Length scaling performance of the best spatial-transfer model under different data budgets. All evaluations are conducted on holdout nodes within the training map (i.e., without spatial transfer). Despite variation in budgets, success rate (SR) consistently deteriorates as path length exceeds the training regime, showing that length scaling fails universally.

## D.6 Additional results for length scaling

**Implementation details and analysis of Figure 1.** We report the length scaling performance of the strongest spatial-transfer model identified in Section 4 (high-budget setting with all budget allocated to questions and high primitive coverage–diversity). Results for other budgets are provided in Section D.6.1.

We measure success rate (SR) on both holdout nodes within the training map (*No spatial transfer*) and spatially disjoint maps (*Spatial transfer*). For each length group, evaluation is conducted on 3,000 randomly sampled unseen node pairs.

The trends are nearly identical across the two settings: while the model achieves near-perfect generalization within the training length regime (blue region), SR rapidly deteriorates once path length exceeds the training maximum (red region). This confirms that **even when spatial transfer succeeds, length scaling can fail**.

### D.6.1 Length scaling performance under different budgets

For completeness, we also evaluate length scaling across different data budgets (Figure 14). For each budget, we select the best-performing spatial-transfer model and report success rates (SR) on holdout node pairs with longer paths between them within the training map.

### D.6.2 Length scaling under a relaxed feasibility metric

We additionally evaluate navigation under a relaxed metric, valid rates, that counts any trajectory reaching the goal (without using invalid edges) as correct, rather than requiring shortest-path optimality. As shown in Figure 15, feasibility remains near perfect for in-distribution lengths but still degrades substantially for longer paths. Relaxing the objective, therefore, does not remove the length-scaling failure. Instead, the higher feasible-path rates relative to shortest-path success suggest that the drop in performance arises from a combination of producing invalid trajectories and producing valid but non-optimal (non-shortest) ones.

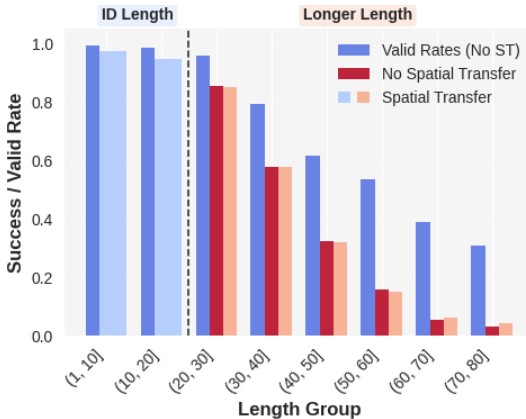

Figure 15: Valid-path rates across length groups. Although absolute performance improves, the same length-scaling failure persists.

## D.7 RL PERFORMANCE FOR MORE TRAINING STEPS

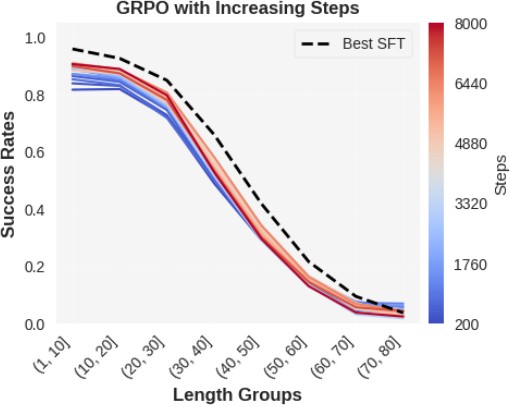

Figure 16: Length scaling for RL under extended training for 20 epochs (1 epoch ≈ 400 steps).

## E PRACTICAL IMPLICATION FROM MATH DOMAIN

To assess the practical relevance of our findings beyond the synthetic map-navigation setting, we conduct a complementary study in the math domain. Specifically, we aim to examine whether, in a more realistic setting, *seeing more questions* remains more impactful than *seeing more solutions*, and whether a question's *coverage* plays a more dominant role than its *diversity*.

### E.1 SETUP

**Dataset** We select the **MathQA** dataset (Amini et al., 2019) for evaluation because: (1) it contains six well-separated conceptual categories (gain, geometry, probability, physics, general, other), spanning a range of difficulties and posing greater challenges to commonly used 7B models than simpler math benchmarks such as GSM8K (Qwen et al., 2025); and (2) critically, it provides *linearized operation programs*. These programs provide a direct way to extract the primitives within questions.

For instance, the problem:

"The ratio between the length and breadth of a rectangular park is $3 : 2$. A man cycles around the boundary at 12 km/hr and completes one round in 8 minutes. What is the area of the park?"

has the corresponding operation program:

divide(12, 60) | multiply(#0, 8) | multiply(#1, 1000) | add(3, 2) | multiply(2, #3) | divide(#2, #4) | multiply(#5, 3) | multiply(#5, 2) | rectangle_area(#6, #7)

Each operation program can be converted into an unordered multiset of operations, which captures the reusable element set required by the problem (i.e., its primitive identity). This representation allows us to define both *coverage* and *diversity* directly on mathematical problems. For datasets without human-provided formulas, we verify that a modern LLM can reliably extract the underlying operation sets from natural-language questions using a short instruction prompt (e.g., `["compute rectangle area", "multiply", ...]`). A sample prompt–response pair is provided in Section F.1.

**Definitions of terms** We restate the core terms (questions, solutions, coverage, diversity) in the context of this math setting:

- **Questions:** Each distinct math word problem.
- **Solutions:** For each question, multiple high-quality reasoning traces may exist, and each trace is counted as a solution. We use DeepSeek-R1 (DeepSeek-AI et al., 2025) to produce such traces (i.e., explicit chain-of-thought outputs in the form of "Let's think step-by-step.").
- **Coverage and diversity in the math domain.** Each MathQA problem is paired with a linearized operation program specifying the sequence of mathematical operations used to solve the problem. To align these programs with the coverage–diversity framework introduced in our map-navigation setting, we decompose them into two orthogonal components:
  - **Coverage.** A single operation (e.g., `add`, `multiply`) is too coarse to characterize mathematical problem types, as most problems reuse the same small set of basic operations. What distinguishes one problem type from another is the *combination* of operations required. Therefore, we treat the *primitive operation-set*—the unordered multiset of operations appearing in an operation program—as the atomic semantic unit of a mathematical problem. This operation set captures the underlying conceptual skill and serves as the minimal distinguishing signature of a problem type. Under this abstraction, coverage measures how many distinct operation-sets the model encounters during training.
  - **Diversity.** In the map setting, diversity measures how many distinct composition patterns exist under the same primitive support—that is, how flexibly a primitive participates in different relational structures. In the math domain, the analogous notion remains: *the number of distinct program structures (operator orderings) that instantiate the same primitive operation-set*. This measures how flexibly a fixed operation set can be composed into different reasoning chains, without introducing new skills.

  For example, the two operation programs below share the same primitive operation-set but differ in ordering; thus, they contribute to diversity but not coverage:

  [divide, multiply, add, rectangle_area]

  [add, divide, multiply, rectangle_area]

We fine-tune Qwen2.5-7B-Instruct (Team, 2024) across three representative difficulty categories in MathQA: `probability` (easy), `gain` (medium), and `physics` (hard). For each category, we fix a tight training budget of roughly $20\%$ of its available samples (approximately 1,000 examples), except for the `probability` split, which uses $50\%$ due to its extremely small size. All models are evaluated on the test set corresponding to the same category. As described above, we use DeepSeek-R1 to generate high-quality chain-of-thought traces for each question and construct three training regimes:

Table 5: Performance of different data regimes across three MathQA categories.

|  |  | probability | gain | physics |
|---|---|---|---|---|
| **Qwen2.5-7B-Instruct** | – | 0.729 | 0.70 | 0.68 |
| **More questions** | High Coverage | **0.792** | **0.82** | **0.77** |
|  | High Diversity | 0.792 | 0.74 | 0.74 |
| **More solutions** | – | 0.771 | 0.72 | 0.70 |

- **More Questions:** each question is paired with exactly *one* solution, enabling a larger number of distinct questions to be included under the same training budget. This includes two cases:
  - **High Coverage:** we include as many distinct primitive operation-sets as possible, resulting $n$ questions per set;
  - **High Diversity:** for each operation-set we include $10n$ distinct questions, which increases structural diversity but necessarily reduces the number of covered operation-sets under the same training budget;
- **More Solutions:** each question is paired with ten independently generated solutions;

### E.2 RESULTS ANALYSIS

The results in Table 5 demonstrate that the core principles identified in our controlled navigation setting apply to the MathQA domain, even though these practical tasks contain heterogeneous natural-language formulations and lack clearly separable generalization axes (e.g., spatial or length extrapolation).

**More questions consistently outperform more solutions.** Across all three categories (i.e., `gain`, `probability`, and `physics`), both *More Questions* regimes (High Coverage and High Diversity) achieve better generalization than *More Solutions*. Notably, these improvements appear under an extremely small training budget: roughly 1,000 samples for category `gain` and `physics`, and only $\sim 200$ for `probability`). Despite such limited supervision, allocating more budget to distinct questions still produces clear performance gains. For instance, in the `gain` category, accuracy rises from 0.70 to 0.82 under `High Coverage`, and a similar increase appears in the harder `physics` category ($0.68 \rightarrow 0.77$).

**Coverage (operation-set variety) remains the dominant factor.** Within the *More Questions* groups, `High Coverage` consistently outperforms `High Diversity` (e.g., 0.82 vs. 0.74 in `gain`, 0.77 vs. 0.74 in `physics`). This suggests that the model gains more from being exposed to a broader set of conceptual skills than from seeing many different applications, orderings, or compositional variants of the same skill set. This echoes the threshold behavior observed in the navigation setting: once the model has seen the *right set of conceptual skills*, generalization improves sharply—even without extensive practice on each skill.

Taken together, these findings reinforce a simple intuition: *under realistic data budgets, breadth matters more than depth.*

## F PROMPTS USED IN THE PIPELINE

### F.1 PROMPT FOR EXTRACTING PRIMITIVE OPERATIONS FROM WORD PROBLEMS

For natural-language math questions without human annotation, we can directly prompt an LLM to extract the underlying primitive operations. Below we provide an example prompt–response pair. The extracted operations are highly reasonable—often comparable to, or even clearer than, the human-provided operation programs used in Section 4.3.

```
Prompt
You are an expert at breaking down math word problems into primitive
mathematical operations.
Given a math word problem, output only a JSON list of short operation
verbs (1-3 words) that describe the steps needed to solve it, such as:
"add", "multiply", "divide", "use ratio", "convert units", "compute
area".
List them in the order they would be used.
Do NOT show explanations or numbers.
Output only the JSON list.
```

```
Model Response
["convert minutes to hours", "multiply", "divide", "use ratio",
"multiply", "compute rectangle area"]
```

### F.2 PROMPT FOR COLLECTING CHAIN-OF-THOUGHT REASONING TRACES

To obtain high-quality chain-of-thought reasoning traces for each math question, we prompt a stronger LLM (DeepSeek-R1 in our implementation) with an instruction that encourages explicit step-by-step reasoning followed by a clearly formatted final answer. Below we provide the exact templates used in our data construction pipeline.

```
System Prompt
You are a helpful math tutor who explains things step-by-step and
always finishes with a clearly formatted final answer.
```

```
User Prompt
Break down your reasoning process step by step, and show your thought
process explicitly.
Separate each step using \n\n.

At the end, conclude with a single line in the exact format:
The answer choice is:  <option>.

Now solve the following multiple-choice math problem:

[Question]
{question_text}
[Solution]
```

### F.3 PROMPT FOR QWEN2.5-7B-INSTRUCT

We use the same prompt for finetuning and evaluation of Qwen2.5-7B-Instruct and our finetuned variants. In all cases, the model is instructed to first produce a step-by-step reasoning trace and then output a clearly formatted final answer, as shown below.

```
System Prompt
You are a helpful assistant.
```

```
User Prompt
Break down your reasoning process step by step, and show your thought
process explicitly.
Separate each step with \n\n.
Conclude with a single line in the exact format:
The answer choice is:  [insert answer choice].

[Question]
{question_text}
[Solution]
```

**Chat template.** The human-readable prompts described above correspond to the `system` and `user` messages used during both fine-tuning and evaluation. In practice, all messages are serialized

Table 6: Error-type statistics for SFT and GRPO across length groups.

| Length Group | Method | Non-Shortest | Not Reach | Invalid Move |
|---|---|---|---|---|
| (10, 20) | SFT | 80.0% | 20.0% | 0% |
| (10, 20) | GRPO | 88.9% | 11.1% | 0% |
| (40, 50) | SFT | 45.0% | 49.0% | 6.0% |
| (40, 50) | GRPO | 43.0% | 50.0% | 7.0% |

using Qwen2.5's official tokenizer chat template (via `apply_chat_template` function). This ensures that both fine-tuning and evaluation use the exact prompt format expected by Qwen2.5-7B-Instruct models, including all special tokens (e.g., `<|im_start|>`) and role indicators required by the tokenizer.

## G  QUALITATIVE ANALYSIS OF NAVIGATION FAILURE CASES

To complement the quantitative results in the main text, we provide qualitative examples and a systematic summary of failure modes for both SFT (coverage $= 0.6$, diversity $= 64$, and maximizing the number of questions) and the corresponding GRPO model (16 rollouts) across two representative length groups: **(10, 20)** (within the training-length regime) and **(40, 50)** (longer length regime). The model's prediction errors consistently fall into the following three categories; we did not observe additional or unexpected behaviors (e.g., producing no trajectory or starting from the wrong initial node):

- **Valid but non-shortest path**
- **Did not reach target**
- **Invalid move**

Table 6 summarizes error statistics. Representative visualizations are presented in Figures Figures 17 to 20.

These qualitative findings show that SFT and GRPO exhibit nearly identical failure modes, reinforcing our conclusion that GRPO stabilizes training but does not surpass the performance ceiling established by the best SFT model.

Figure 17: Representative failure cases (SFT) for the (10, 20) length group.

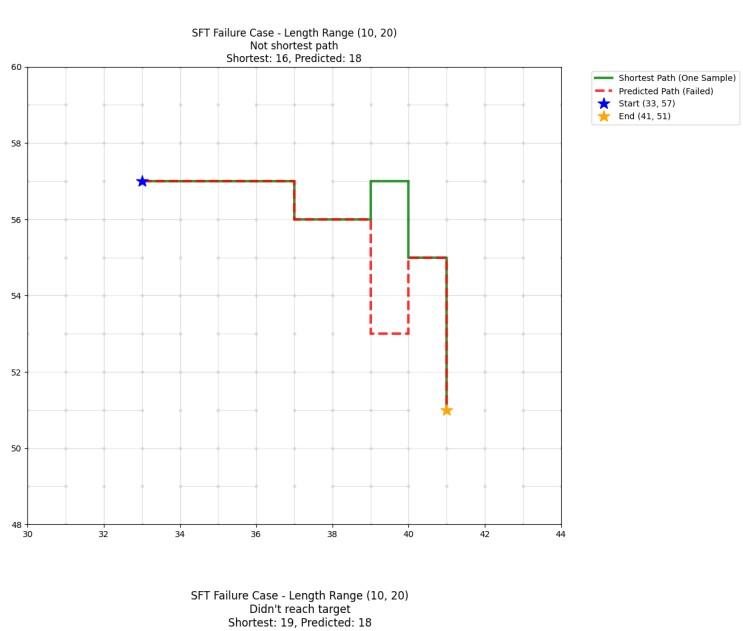

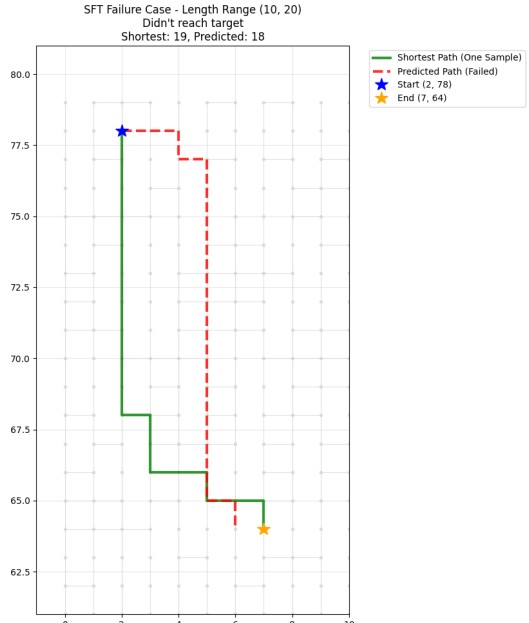

Figure 18: Representative failure cases (GRPO) for the (10, 20) length group.

Figure 19: Representative failure cases (SFT) for the (40, 50) length group.

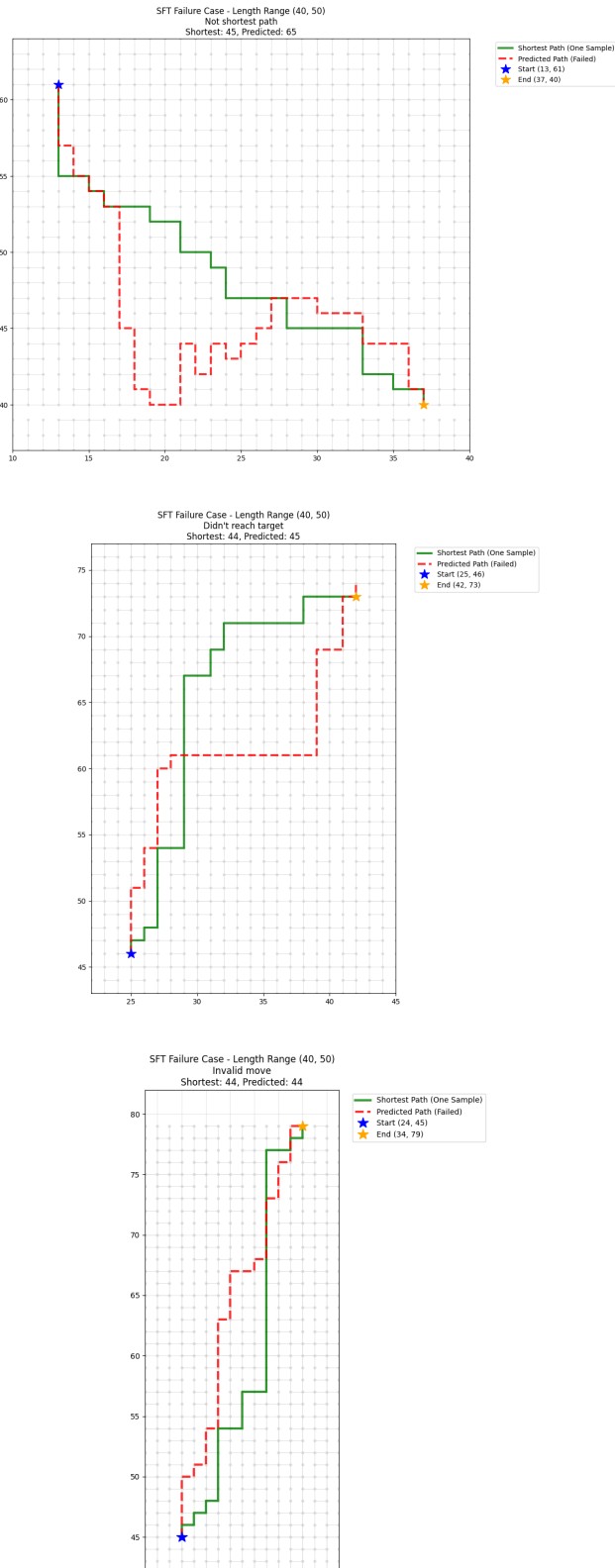

Figure 20: Representative failure cases (GRPO) for the (40, 50) length group.

