# OpenReview forum: "Generalization in LLM Problem Solving: The Case of the Shortest Path"
_ICLR.cc/2026/Conference — ICLR 2026 Poster_

### Official Review · Reviewer_QYZU · 2025-10-28

**Soundness:** 2
**Presentation:** 1
**Contribution:** 1
**Rating:** 2
**Confidence:** 4

**Summary:**

This paper introduces a map navigation benchmark based on 2D sparse grid navigation tasks, and uses it to evaluate transformer models with LLaMA-style architecture,
The authors analyze model performance on this benchmark using two dimensions:
- Transfer (systematicity): solving the same class of problems in new environments, and
- Scaling (productivity): solving harder problems after mastering simpler ones.
They consider two model training paradigms: (1) Supervised fine-tuning, with paths encoded as movement directions; model predicts continuation given start/end prompt. (2) Reinforcement learning, giving binary reward for valid shortest paths. They find that for supervised fine-tuning distinct questions drive transfer more than multiple solutions per question.
To determine which types of questions best support spatial transfer they focus on coverage (fraction of unique nodes in the training map included in training) vs diversity (average number of distinct endpoints each start node is paired with.) of training questions.

**Strengths:**

The authors focus on defined generalization dimensions (transfer and scaling) and the experimental design appears to be sound, separating coverage vs diversity.

Overall, this work clearly *intended as* a methodologically solid framework with careful experiments, however poor presentation makes this conceptual design very hard to understand and evaluate.

**Weaknesses:**

Presentation is very poor, making it hard to follow this paper.

Beginning as early as the abstract, the writing is often superficially coherent, but conceptually incoherent - reflecting perhaps the lack of conceptual depth, or perhaps a last minute writing of work that is intended as well thought out, but obscured by poor presentation. For instance:

"_Someone who learns to walk shortest paths in New York can, upon receiving a map of Paris, immediately apply the same rule to navigate, despite never practicing there._"

Same rule (as in, the same heuristic) or same algorithm (as in, Djkstra's algorithm)?
Much of the field would not agree with this statement, as New York has a grid-like North American design, and Paris is an older city - meaning that different rules/heuristics and approximations will be optimal.

The next sentence then jumps topic to combining rules - "_This ability to recombine known rules to solve novel problems exemplifies compositional generalization_.."
There is no sense how this is related to the previous sentence, or the remainder of the abstract.

I am unable to evaluate the paper, because I can not unearth the content from the ineffective presentation. I feel that this work is not yet ready for an archival publication, but after a careful revision it might be.

**Questions:**

N/A

---

### Official Review · Reviewer_QLfE · 2025-11-01

**Soundness:** 3
**Presentation:** 2
**Contribution:** 3
**Rating:** 6
**Confidence:** 3

**Summary:**

In this paper, the authors focus on the understanding of the generalization and problem solving abilities of state of the art machine learning models. The authors use the simple example of token based map navigation to study these questions. To this end, they explore a series of highly constrained evaluations in order to disentangle the effects of data coverage, diversity, and scaling.In addition, they look at the effects of reinforcement learning. A number of key findings are uncovered. Coverage is very important for spatial transfe, with diversity generally less important. Reinforcement improves optimization but does not improve transfer.

**Strengths:**

1. Generalization, and the understanding of what leads to generalization, is of fundamental importance in machine learning.

2. Paper is well written and easy to understand.

3. Although the problem and domain is quite narrow (token based navigation), evaluation is quite detailed and thorough, and the results are of interest to the community.

**Weaknesses:**

1. The domain used is a toy problem (token based navigation) and very narrow. It is unclear if the results in the paper hold in other domains or problems in machine learning.

2. Qualitative results or a visualization of navigation capabilities (or failure cases) would strengthen the paper.

**Questions:**

1. Could the authors please elaborate on how the results could apply to other domains? The test-bed of navigation is quite narrow.

2. Would it be possible to include any qualitative examples or failure cases to understand the difference in performance between various regimes?

---

> ### Author Response · Authors · 2025-11-27
> **To Reviewer QLfE**
>
> Thank you for your valuable suggestions and constructive feedback. They greatly help improve the quality of our paper. Since each question is directly tied to its corresponding weakness, we address them in pairs (W1–Q1 and W2–Q2) below.
>
> > ***W1: The domain used is a toy problem (token based navigation) and very narrow. It is unclear if the results in the paper hold in other domains or problems in machine learning. (Q1: Could the authors please elaborate on how the results could apply to other domains? The test-bed of navigation is quite narrow.)***
>
> Thanks for the intriguing question. We agree that demonstrating applicability to domains such as math or coding would further strengthen our results. Our synthetic navigation environment is deliberately designed as a controlled setting where coverage, diversity, and length can be varied independently—something that is difficult to achieve in open-domain tasks. The goal is not to position navigation as a target application domain, but to isolate structural factors that affect extrapolation.
>
> To examine whether these results extend beyond navigation, we **added a new experiment on MathQA**, a natural-language math benchmark with diverse task categories and difficulty levels, using the **Qwen2.5-7B-Instruct** model.
>
> We define ***coverage*** as the number of distinct operation-sets (conceptual skill sets) present in training, and ***diversity*** as the number of different instantiations or orderings of an operation-set (see **Appendix D** for details). We use DeepSeek-R1 to generate high-quality **chain-of-thought traces (i.e., solutions)**, and we set ***More Questions*** to use one solution per question and ***More Solutions*** to use ten solutions per question (adjusting the number of questions accordingly to keep the overall training budget fixed).
>
> ***Results:*** (Full details of these experiments are provided in Appendix D, with a case study version in Section 3.4.)
> |                           |               | probability | gain | physics |
> |---------------------------|---------------|-------------|------|---------|
> | **Qwen2.5-7B-Instruct (before training)** | --            | 0.729       | 0.70 | 0.68    |
> | **More questions**        | High Coverage | **0.792**       | **0.82** | **0.77**    |
> |                           | High Diversity| 0.792       | 0.74 | 0.74    |
> | **More solutions**        | --            | 0.771       | 0.72 | 0.70    |
>
> Across experiments spanning three math categories (probability, gain, physics), the key takeaways from our controlled study carry over:
> - Spending the budget on exposing the model to **more distinct questions is more beneficial than** allocating the budget to providing **more diverse solutions**, and
> - **Coverage (the variety of operation-sets present in the questions) is the dominant factor**
>
> These results demonstrate that the phenomena we study are not specific to token-based navigation, but also manifest in more realistic math domains and larger-scale LLMs with natural-language inputs. Synthetic and real-world tasks thus play complementary roles: the former allows controlled causal analysis, while the latter confirms that the identified mechanisms extend beyond the toy setting.
>
> > ***W2: Qualitative results or a visualization of navigation capabilities (or failure cases) would strengthen the paper. (Q2: Would it be possible to include any qualitative examples or failure cases to understand the difference in performance between various regimes?)***
>
> Thank you for this helpful suggestion! Qualitative examples indeed provide additional insight into model behavior. In **Appendix F** of the revised manuscript, we have added a dedicated section that (i) reports detailed statistics characterizing the failure behaviors of the SFT model (coverage = 0.6, diversity = 64) and its corresponding GRPO variant under both the within-training-length regime (10–20) and the longer-length regime (40–50), and (ii) presents visualizations of representative failure-case trajectories for each error type. These examples cover all failure modes we observed, including valid-but-non-shortest paths, failures to reach the target, and invalid moves.
>
> The qualitative patterns closely mirror the quantitative findings. In particular, **SFT and GRPO exhibit identical failure modes and nearly identical error distributions across failure modes**, reinforcing our main conclusion that GRPO stabilizes training but does not surpass the extrapolation performance of the best SFT model. We hope these additions help make the model’s behavior clearer and more interpretable to the reader.

---

### Official Review · Reviewer_7hks · 2025-11-01

**Soundness:** 3
**Presentation:** 3
**Contribution:** 3
**Rating:** 6
**Confidence:** 4

**Summary:**

This paper introduces a controlled “map worlds” testbed to dissect extrapolative problem solving in Transformers along two axes of compositional generalization: spatial transfer (applying rules to disjoint environments) and length scaling (solving longer instances). Using 8-layer LLaMA-style models pretrained on random walks and fine-tuned (or RL-trained) to generate shortest paths, the authors show: (1) under fixed budget, allocating data to many distinct questions (start–end pairs) beats collecting many solutions per question; (2) coverage of primitives sets the transfer ceiling, unlocked by modest diversity, with a sharp SR inflection around 20–25% coverage; (3) diversity yields log-linear gains at moderate/high coverage but can hurt at low coverage; (4) length scaling generally fails unless training includes a small curriculum of neighboring-but-longer examples (+2–4 steps), while shorter or much longer examples help little or harm; and (5) RL (Dr.GRPO) stabilizes training and avoids overfitting but never exceeds the best SFT performance. The work offers practical data-design guidance (prioritize coverage and distinct questions; minimal solution diversity) and clarifies RL’s role as stabilizer rather than capability expander in this setting.

**Strengths:**

- 1. Clear decomposition of generalization: The paper cleanly separates spatial transfer from length scaling in a controlled “map worlds” testbed, enabling precise causal insights that are hard to obtain in natural language settings.

- 2. Controllable data design: The graph-based framework enables precise control over dataset factors, including node coverage (fraction of unique nodes appearing in training questions), pairing diversity (average number of endpoints per start node), and path-length regimes (via sampling constraints). This controllability supports principled ablations on how dataset composition, independent of total record count, shapes generalization.

- 3.The paper systematically varies training budget as a percentage of the possible start–end records and studies trade-offs between the number of distinct questions and the number of solutions per question under approximately fixed total records. These ablations quantify data-efficiency regimes and clarify which allocations (e.g., higher question diversity vs. multiple solutions per question) most improve spatial transfer, though matching total optimization steps (compute) is not explicitly controlled.

**Weaknesses:**

- Synthetic scope and small models: Results are demonstrated on small LLaMA-style models and synthetic sparse grids. External validity to larger models, richer graph families, or open-domain language tasks is uncertain.
- RL ceiling bounded by SFT without diagnosis: RL consistently fails to surpass the best SFT model, but the paper provides limited analysis of why (e.g., exploration limits, reward sparsity, credit assignment), and does not test alternative rewards (e.g., feasible-path rewards) or curricula beyond near-boundary lengths.

**Questions:**

- 1. **Pre-training**: Recent work suggests[1,2] that continued or mid-stage pretraining can substantially improve downstream RL gains, so it’s reasonable to worry that the findings here may be shaped by an underpowered pretraining phase. The paper pretrains on long random walks to impart “map semantics” and confirms this doesn’t leak shortest-path skills, but it does not report scale, steps, or convergence, nor ablate the pretraining budget. Without those controls, we can’t tell whether stronger pretraining would raise spatial-transfer ceilings, improve length scaling, or allow RL to surpass SFT.

- 2. **Compute**: As mentioned in the paper that the budget is controlled by the percentage of possible records. Are experiments controlled with same amount of training compute? i.e. are experiments with fewer training data training on more epochs to match equal optimization steps?

- 3. **Length statistic details**: The study carefully separates spatial transfer (within training-length regime) from length scaling (strictly longer than training), but it does not provide the actual path-length distributions for training or evaluation, nor how those distributions shift as coverage increases. Reporting histograms (x: path length, y: sample count) for each training setup and test split, along with Lmax per setup and the fraction of samples near Lmax, would reveal whether performance changes stem from coverage/diversity design or from incidental shifts in length exposure.

- 4. **Long-to-short generalization**: The paper analyzes “short-to-long” length scaling but does not test the reverse direction. A simple control would train on a length band with relatively high minimum length and evaluate on shorter paths to see whether a long-to-short generalization effect exists.

- 5. **Non-optimal navigation evaluation**: In some cases of real-world navigation tasks, people may only care about finding a feasible path rather than shortest. Is there evaluation results for this setting? Will generalization happen when training on navigating shortest paths while evaluating feasible paths?

[1] Wang, Zengzhi, Fan Zhou, Xuefeng Li, and Pengfei Liu. "Octothinker: Mid-training incentivizes reinforcement learning scaling." arXiv preprint arXiv:2506.20512 (2025).

[2] Cen, Zhepeng, Yihang Yao, William Han, Zuxin Liu, and Ding Zhao. "Behavior Injection: Preparing Language Models for Reinforcement Learning." arXiv preprint arXiv:2505.18917 (2025).

---

> ### Author Response · Authors · 2025-11-27
> **(1/4) To Reviewer 7hks**
>
> Thank you for your supportive review and constructive feedback. We respond to each point below.
>
> > ***W1: Synthetic scope and small models:  Results are demonstrated on small LLaMA-style models and synthetic sparse grids. External validity to larger models, richer graph families, or open-domain language tasks is uncertain.***
>
> We appreciate the reviewer’s concern. While our core experiments use small LLaMA-style models in synthetic sparse-grid environments to allow clean control over coverage, diversity, and length, we have added a new experiment on math domain tasks using the substantially larger Qwen2.5-7B-Instruct model to test whether the same patterns hold at a different scale model and in a domain far removed from grid navigation.
>
> **MathQA experiment (newly added).**
>
>  MathQA is a natural-language math dataset with diverse questions spanning multiple categories and annotated linearized operation programs. These annotations allow us to define, for each problem:
>  (i) **coverage**: the number of distinct operation-sets (conceptual skill sets) seen during training, and
>  (ii) **diversity**: the number of different combinations or orderings within a skill set.
> These definitions parallel the concepts in our synthetic testbed while applying them in a natural-language setting. *We also verify that LLMs can extract operation-sets directly from text (**Appendix D–E**), enabling analyses of other math datasets even when formulas are not provided.*
> To compare **more questions** versus **more solutions** under a fixed budget, we generate high-quality solutions (i.e., chain-of-thought traces) using DeepSeek-R1. We evaluate three representative MathQA categories of varying difficulty (probability/gain/physics).
>
> **Results.** (Additional details appear in **Appendix D**, and a concise case study is provided in **Section 3.4**.)
>
> |                           |               | probability | gain | physics |
> |---------------------------|---------------|-------------|------|---------|
> | **Qwen2.5-7B-Instruct (before training)** | --            | 0.729       | 0.70 | 0.68    |
> | **More questions**        | High Coverage | **0.792**       | **0.82** | **0.77**    |
> |                           | High Diversity| 0.792       | 0.74 | 0.74    |
> | **More solutions**        | --            | 0.771       | 0.72 | 0.70    |
>
> Despite the increased model size and domain complexity, the key takeaways from our findings continue to hold across all three difficulty categories. Specifically:
>  - (1) training on **more distinct questions consistently outperforms** providing **more solutions** per question, and
>  - (2) **coverage (operation-set variety here) remains the dominant factor**—even under extremely limited supervision ($\sim$1k samples).
>
> The trends observed in our controlled setting thus remain clearly visible in this more realistic evaluation.
>
> At the same time, we emphasize that our goal is to build a clean and controllable testbed to isolate structural generalization factors—something that is difficult to do directly in open-domain LLMs. Our synthetic environment lets us vary primitives, adjacency patterns, and length in ways that are not confounded by world knowledge or pretraining priors. The MathQA results complement this by showing that the discovered mechanisms persist when moving to larger models and natural-language reasoning tasks.

---

> ### Author Response · Authors · 2025-11-27
> **(2/4) To Reviewer 7hks**
>
> > ***W2: RL ceiling bounded by SFT without diagnosis: RL consistently fails to surpass the best SFT model, but the paper provides limited analysis of why (e.g., exploration limits, reward sparsity, credit assignment), and does not test alternative rewards (e.g., feasible-path rewards) or curricula beyond near-boundary lengths.***
>
> We understand the reviewer’s concern that the RL ceiling might be due to a suboptimal RL setup and thus require deeper diagnosis. However, our findings are consistent with a growing body of recent work [1,2,3] showing that when RL and SFT optimize the *same* underlying objective, RL should **not** be expected to outperform SFT. **Our results provide direct support for the theoretical results in [1]**, which argues that RL appears stronger in many domains primarily due to the **generation–verification gap**: generating optimal solutions is difficult, but verifying them is easy. Therefore, RL’s exploration and rejection sampling allow it to surface solutions that SFT does not encounter. In our synthetic setting, the optimal policy is explicit, so **generating optimal actions is no longer the bottleneck**. Once the generation–verification gap is removed, RL loses its typical advantage, and we observe that RL does not surpass SFT.
>
> A detailed analysis of RL’s optimization dynamics (e.g., identifying which algorithmic factors could improve performance) is beyond the scope of this work and remains an open problem in RL-for-reasoning research. However, to better understand the bounded behavior of RL in our setting, we added a qualitative failure-case analysis in **Appendix F**. The statistics show that **RL and SFT exhibit the same failure modes with nearly identical error distributions**. This naturally reminds us of the findings in [3], which show that the reasoning traces produced by an RL-tuned model already exist in the base model. In real practice, the “base model’’ used for RL is typically already strengthened by substantial SFT and thus has strong task-specific reasoning ability. In our setting, this role is played by the SFT model. Consequently, once the SFT data are sufficient to close the generation–verification gap, RL neither improves extrapolation nor avoids the errors already made by SFT.
>
> ***Regarding alternative reward designs:*** we focus on outcome-based rewards because they are the standard choice in recent RL-for-reasoning setups. We experimented with other reward designs, such as progress-based shaping, but observed clear **reward hacking** behaviors (e.g., collapsing to short yet invalid paths). This is consistent with prior findings [4,5], which show that proxy rewards misaligned with the true task objective tend to induce reward hacking, whereas verification-based rewards directly encode the goal and are therefore less vulnerable to such failures.
>
> Finally, “feasible-path rewards” are already incorporated in our formulation, since the task objective is to generate a **feasible shortest path**. Rewards that encourage feasibility only without enforcing optimality introduce misaligned objectives, and therefore do not help with learning.
>
> *[1] Swamy, G., Choudhury, S., Sun, W., Wu, Z. S., & Bagnell, J. A. (2025). All roads lead to likelihood: The value of reinforcement learning in fine-tuning.*
>
> *[2] Ma, L., Liang, H., Qiang, M., Tang, L., Ma, X., Wong, Z. H., ... & Zhang, W. (2025). Learning What Reinforcement Learning Can't: Interleaved Online Fine-Tuning for Hardest Questions.*
>
> *[3] Yue, Y., Chen, Z., Lu, R., Zhao, A., Wang, Z., Song, S., & Huang, G. (2025). Does reinforcement learning really incentivize reasoning capacity in llms beyond the base model?*
>
> *[6] Perez, E., Ringer, S., Lukosiute, K., Nguyen, K., Chen, E., Heiner, S., ... & Kaplan, J. (2023, July). Discovering language model behaviors with model-written evaluations.*
>
> *[7] Tarek, M. F. B., & Beheshti, R. (2025). Reward hacking mitigation using verifiable composite rewards*

---

> ### Author Response · Authors · 2025-11-27
> **(3/4) To Reviewer 7hks**
>
> > ***Q1: Pre-training: Recent work suggests[1,2] that continued or mid-stage pretraining can substantially improve downstream RL gains, so it’s reasonable to worry that the findings here may be shaped by an underpowered pretraining phase. The paper pretrains on long random walks to impart “map semantics” and confirms this doesn’t leak shortest-path skills, but it does not report scale, steps, or convergence, nor ablate the pretraining budget. Without those controls, we can’t tell whether stronger pretraining would raise spatial-transfer ceilings, improve length scaling, or allow RL to surpass SFT.***
>
> Thank you for raising this point. We have added the specifications of the pretraining stage in Appendix C.1. Our experiments use a substantial pretraining budget: **10M random-walk trajectories ($\approx$ 1.3B tokens)** trained for **124,999 steps**, which fully saturates the pretraining objective. Because the objective is random-walk prediction rather than a task with a clear semantic target, **standard convergence metrics (e.g., validation loss plateau) are not very informative**. Instead, we monitored a structural proxy—the model’s random-walk **valid-path rate**, i.e., the probability that it generates a valid walk when rolled out from its own predictions.
>
> As shown in Table 3, **the pretrained model achieves a valid-path rate of 1.0, indicating that pretraining has sufficiently captured the intended “map semantics’’** while retaining zero shortest-path capability. We also note that the choice of 10M is not arbitrary but is **based on this convergence signal**: smaller pretraining corpora yield slightly lower valid-path rates (e.g., $\approx$ 0.97 at 5M trajectories, corresponding to 65,105 steps), and the rate saturates after around 100k steps, so we selected the 10M setting. Since increasing the pretraining budget further would not introduce any information correlated with shortest-path reasoning, we believe the downstream results are not affected by pretraining capacity.
>
> > ***Q2: Compute: As mentioned in the paper that the budget is controlled by the percentage of possible records. Are experiments controlled with same amount of training compute? i.e. are experiments with fewer training data training on more epochs to match equal optimization steps?***
>
> Yes, all experiments control for training compute (by fixing the total number of training samples rather than steps).
> - Question vs. solution experiments: We fix compute by ensuring that #questions × average #solutions per question equals a constant total budget (Lines 171–173).
> - Coverage vs. diversity in questions experiments: We fix the total number of records by varying the number of solutions per question (Line 256). We didn't match by epochs because that would oversample the minimum-coverage-diversity setting by 2560$\times$, which would force extreme overfitting. However, while this setup is asymmetric, it actually strengthens the conclusion: higher coverage performs better despite having fewer effective samples, reinforcing that coverage matters more than diversity (as discussed in Line 320–323).
>
> > ***Q3: Length statistic details: The study carefully separates spatial transfer (within training-length regime) from length scaling (strictly longer than training), but it does not provide the actual path-length distributions for training or evaluation, nor how those distributions shift as coverage increases. Reporting histograms (x: path length, y: sample count) for each training setup and test split, along with Lmax per setup and the fraction of samples near Lmax, would reveal whether performance changes stem from coverage/diversity design or from incidental shifts in length exposure.***
>
> Thank you for the suggestion. We have added histograms of the shortest-path length distributions under increasing coverage (with fixed diversity); the results are shown in **Figure 10 in Appendix C.4** in the revised version. As the distributions are nearly identical across all coverage values, we did not additionally plot $L_{\max}$ or the fraction of samples near $L_{\max}$.
>
> Figure 10 shows that the distance distribution remains highly stable across coverage settings (with similar mean and std). This confirms that the performance trends in the main paper are not due to incidental changes in length exposure but arise from the coverage and diversity manipulations themselves.

---

> ### Author Response · Authors · 2025-11-27
> **(4/4) To Reviewer 7hks**
>
> > ***Q4: Long-to-short generalization: The paper analyzes “short-to-long” length scaling but does not test the reverse direction. A simple control would train on a length band with relatively high minimum length and evaluate on shorter paths to see whether a long-to-short generalization effect exists.***
>
> Thank you for the insightful suggestion. Testing **long-to-short generalization** is indeed a useful diagnostic for whether the model is merely relying on short-pattern matching or has learned a more global shortest-path rule. In our original setup, the model is trained on paths of length 1–20. Following the reviewer’s suggestion, we removed all training samples shorter than length 8, as well as all longer trajectories that contain sub-paths between the test nodes (length 1-7), and then evaluated the model on shorter paths. The results are shown below:
>
> ### **Long → Short Generalization Results**
> | Path Length | Success Rate |
> | ----------- | ------------ |
> | 2 | 0.95 |
> | 3 | 0.97 |
> | 4 | 0.94 |
> | 5 | 0.98 |
> | 6 | 0.98 |
> | 7 | 0.99 |
>
> These results indicate that the model generalizes **strongly from longer paths to shorter ones**, suggesting that the difficulty in short-to-long scaling does not stem from the model failing to acquire a global rule. Instead, the failure is asymmetric: **long→short generalization is easy, but short→long extrapolation remains hard**.
>
> > ***Q5: Non-optimal navigation evaluation: In some cases of real-world navigation tasks, people may only care about finding a feasible path rather than shortest. Is there evaluation results for this setting? Will generalization happen when training on navigating shortest paths while evaluating feasible paths?***
>
> Thank you for the suggestion. We added an evaluation under this relaxed valid rates metric that counts any trajectory reaching the goal using valid edges as correct (without requiring shortest-path optimality). The results are included in the revision (**Figure 10 in Appendix C.5**). As expected, although this relaxed metric yields higher absolute success rates, the same length-scaling failure persists. This indicates that performance drops in length generalization arise from both invalid trajectories and valid but non-shortest ones.

---

### Official Review · Reviewer_UaSt · 2025-11-10

**Soundness:** 3
**Presentation:** 3
**Contribution:** 2
**Rating:** 2
**Confidence:** 3

**Summary:**

This paper investigates compositional generalization in neural networks by introducing a controlled map-navigation testbed that cleanly separates two fundamental dimensions: spatial transfer (applying learned rules to entirely new environments) and length scaling (solving longer problems than seen during training). Using shortest-path navigation on 2D grid maps, the authors train small transformer models and systematically study how data selection and training paradigms affect extrapolative problem-solving. Their key findings show that spatial transfer is primarily enabled by maximizing the number of distinct training questions with high primitive coverage and only modest diversity, rather than collecting multiple solutions per question. In contrast, length scaling requires exposure to neighboring-but-longer examples and cannot be achieved through spatial transfer alone. Comparing supervised fine-tuning (SFT) and reinforcement learning (RL), they find that while RL provides training stability and prevents overfitting, it does not unlock capabilities beyond what the best SFT model can achieve—the performance ceiling is always set by SFT when data is sufficient and high-quality. These results provide principled insights into data-efficient training strategies for extrapolative problem-solving in language models.

**Strengths:**

The map-navigation testbed is cleverly constructed to provide truly disjoint test domains while maintaining control over primitives and rules, enabling causal analysis of data properties that is nearly impossible in realistic settings. The systematic experiments on coverage versus diversity provide actionable insights for data collection, with clear resource-efficiency guidelines showing that high coverage with modest diversity outperforms the reverse. The comparison between SFT and RL across both generalization dimensions is thorough and well-controlled, with multiple rollout configurations and warm-start conditions. The writing is clear, findings are well-supported by comprehensive ablations, and the practical takeaways are concrete.

**Weaknesses:**

The main limitation is generalizability. All conclusions are drawn from small (8-layer) transformers on synthetic shortest-path tasks, raising significant questions about whether these insights transfer to billion-parameter LLMs solving complex mathematical or coding problems. The testbed, while controlled, may oversimplify compositional reasoning by reducing it to spatial navigation, potentially missing critical aspects like nested recursion, abstract rule composition, or semantic understanding that characterize real problem-solving. The finding that RL doesn't surpass SFT contradicts recent work showing RL enables extrapolation in mathematical reasoning, though the authors acknowledge this may reflect differences in data quality rather than fundamental capabilities. The paper also lacks theoretical analysis explaining why coverage dominates diversity or why length scaling requires neighboring examples, relying primarily on empirical observations.
Also, I want you to compare offline planning line of works that also tries to enhance performance on maze tasks, and position how your work is better than existing works on offline planning in advance.
Here are some example papers:
Yang et al, Chain of Thought Imitation with Procedure Cloning
Kim et al, How language models extrapolate outside the training data: A case study in Textualized Gridworld

**Questions:**

How sensitive are these findings to model scale? Would coverage versus diversity trade-offs shift dramatically with GPT-4-scale models that have vastly more capacity? Can the neighboring-but-longer insight be formalized into a curriculum learning strategy applicable beyond shortest paths? How do these findings relate to recent work on test-time compute scaling and chain-of-thought reasoning, where models might internally generate longer reasoning traces? Finally, would outcome-based RL (as tested here) versus process-based RL that rewards intermediate steps show different results for length scaling, given that shortest-path generation resembles step-by-step reasoning?

---

> ### Author Response · Authors · 2025-11-27
> **(1/3) To Reviewer UaSt**
>
> We appreciate the reviewer’s detailed review and comments. We respond to each point below.
>
> > ***W1: The main limitation is generalizability. & Q1: How sensitive are these findings to model scale?***
>
> We appreciate the reviewer’s concern, and we agree that synthetic environments inevitably trade realism for controllability. As it essentially overlaps with the first question (*“Q1: How sensitive are these findings to model scale?”*), we address them together below:
>
> To evaluate whether our findings extend beyond the controlled shortest-path setting, we **added a new case study on the MathQA dataset using the Qwen2.5-7B-Instruct model**. MathQA consists of diverse natural-language math questions spanning multiple categories (and hence multiple difficulty levels), and it provides annotated linearized operation programs. These annotations allow us to define for each problem: (i) **coverage**—the number of distinct operation-sets (conceptual skill sets) seen during training, and (ii) **diversity**—the number of different combinations or orderings within the same skill set. We also verify that LLMs can reliably extract operation-sets from raw text (Appendix D–E), meaning similar analyses can be applied even without human-provided formulas. To vary the supervision between **more questions** and **more solutions** under a fixed budget, we use DeepSeek-R1 to generate high-quality *chain-of-thought solutions* for each question.
>
> We evaluate three representative categories: ***probability*** (easy), ***gain*** (medium), and ***physics*** (hard). The ***gain*** and ***physics*** splits contain roughly $1{,}000$ samples each, while ***probability*** contains only $\sim 200$ examples due to its small size. The results are shown below:
>
> |                           |               | probability | gain | physics |
> |---------------------------|---------------|-------------|------|---------|
> | **Qwen2.5-7B-Instruct (before training)** | --            | 0.729       | 0.70 | 0.68    |
> | **More questions**        | High Coverage | **0.792**       | **0.82** | **0.77**    |
> |                           | High Diversity| 0.792       | 0.74 | 0.74    |
> | **More solutions**        | --            | 0.771       | 0.72 | 0.70    |
>
> These results echo our controlled findings in a substantially more realistic setting:
>
> **(1) More Questions $>$ More Solutions:**
>  Across all categories, exposing the model to more distinct questions (high coverage or high diversity) consistently yields larger improvements than adding more solutions to the same question.
>
> **(2) High Coverage $>$ High Diversity:**
>  Broader coverage of conceptual skill sets provides larger gains than increasing the number of compositional variants built from the same skills.
>
> This 7B-scale real-world math experiment provides concrete evidence that the phenomena we observe are not artifacts of small 8-layer transformers, but reflect a more general data property that also appears in larger-scale LLMs solving practical reasoning tasks.
> **We added the full experimental details in Appendix D, and a concise case-study version is included in Section 3.4.**
>
> > ***W2: The testbed, while controlled, may oversimplify compositional reasoning by reducing it to spatial navigation***
>
> The navigation task is intentionally simple. This allows us to vary coverage, diversity, and length **without interference from semantics or linguistic variability**, which are major confounders in real tasks. The model still needs to learn local transitions and compose them over many steps, but we avoid higher-level factors that would blur the effect of the variables we study. We clarify that our testbed is designed to isolate the structural components most relevant to spatial reasoning (i.e., coverage, diversity, and length) and to surface insights specific to these factors. Other forms of compositionality (e.g., rule abstraction) are better examined using their own dedicated and appropriately controlled testbeds.

---

> ### Author Response · Authors · 2025-11-27
> **(2/3) To Reviewer UaSt**
>
> > ***W3: The finding that RL doesn't surpass SFT contradicts recent work showing RL enables extrapolation in mathematical reasoning, though the authors acknowledge this may reflect differences in data quality rather than fundamental capabilities.***
>
> It is an ongoing debate how much RL actually contributes, but **a growing body of recent work [1,2,3] is consistent with our findings**. For example, [1] theoretically shows that when RL and SFT optimize the same underlying objective, RL should not outperform SFT; the empirical gains from RL stem from the **generation–verification gap**, i.e., **in most settings, optimal generation is difficult, but verifying and filtering good completions is much easier**. The experiments in [1], along with recent online and reward-rectified SFT approaches [4,5], further support that: When this gap is closed (e.g., by giving SFT access to high-quality, filtered samples), SFT can match or even surpass RL without using an RL objective.
>
> **Our results provide complementary and more direct support**. In our synthetic setting, the optimal policy is explicit, so generating optimal actions is *no longer* the bottleneck.Once the generation–verification gap is removed, RL loses its typical advantage, and we observe that RL does not surpass SFT.
>
> To further diagnose this phenomenon, we added detailed error-type analyses in **Appendix F**. The failure case statistics show that SFT and its corresponding RL model **not only share the same failure modes but also exhibit nearly identical distributions across these modes**. This mirrors, in a reversed perspective, the findings in [3]  that the reasoning traces produced by an RL-tuned model already exist in the base mode. In real practice, the “base model’’ used for RL is usually already strengthened by substantial SFT and therefore already has strong task-relevant reasoning ability. In our setting, this role is played by the SFT model. As a result, with enough "optimal" data to close the generation–verification gap, RL neither improves extrapolation nor avoids SFT’s errors.
>
> > ***W4: Primarily on empirical observations.***
>
> We agree that our contribution is empirical, but all the factors we studied in this work are motivated by theoretical findings (as introduced in Section 3.1 & 3.2). For example, existing theory suggests that both coverage and diversity can help, but says little about how they interact under a fixed budget or how length exposure should be arranged. Our controlled setting reveals concrete patterns—coverage dominating diversity, and the need for neighboring lengths—that can guide future theoretical work. We clarify this positioning more explicitly in the revision.
>
> > ***W5: Relation to offline planning***
>
> Offline-planning and textualized-gridworld work typically provides the full (small) map or explicit transition rules directly in the prompt, and studies how a planner scales when the world is already known. Their goal is therefore not to construct a clean generalization test: when the map is placed in the prompt, and the model is a pretrained foundation model (e.g., one that has been exposed to graph-theoretic or map-like data), the test distribution inevitably resembles patterns the model has already encountered during training.
>
> Our setting asks a fundamentally different question: **Given only trajectories (without maps), can the model infer the underlying adjacency structure and transition rules without ever being shown the map?**
> The model must reconstruct the structure from traces, rather than apply planning over known dynamics.
>
> This design has two advantages. First, because the model is trained from scratch with the map represented entirely through a large discrete vocabulary that captures nodes and adjacency, we can construct a test world that is genuinely out-of-distribution (by controlling which nodes appear during training). This setup is also closer to natural language: the rules for how tokens should be used are not explicitly provided but must instead be inferred from usage patterns. In contrast, settings where the map is provided as structured input cannot guarantee such a clean generalization boundary.
> Second, it isolates the inductive biases of the model: success requires inferring latent structure, not executing explicit planning instructions. Therefore, our testbed provides a cleaner setting for studying extrapolation.

---

> ### Author Response · Authors · 2025-11-27
> **(3/3) To Reviewer UaSt**
>
> > ***Q2: Can the neighboring-but-longer insight be formalized into a curriculum learning strategy applicable beyond shortest paths?***
>
> Our finding is empirical (and we do not claim a universal curriculum rule), but it suggests a simple intuition: models extrapolate more reliably when the training data maintains local continuity along a difficulty axis (here, path length). We view this as a high-level guideline rather than a formal curriculum strategy. Similar continuity may correspond to proof length in mathematics, program size in code generation, or CoT step count in reasoning tasks. Overall, we agree with this perspective, and this intuition is indeed informing some of our ongoing follow-up work.
>
> > ***Q3: How do these findings relate to recent work on test-time compute scaling and chain-of-thought reasoning, where models might internally generate longer reasoning traces?***
>
> This line of work is orthogonal to our setting. In our case, our experiments indicate that extrapolation failures are primarily driven by missing training signals (broad coverage for transfer and neighboring-but-longer examples for scaling), rather than by insufficient test-time reasoning depth. Our task is intentionally simple and direct, which greatly reduces the ambiguity that often arises in practical scenarios about whether additional test-time compute is the main bottleneck. Here, the path itself already provides a direct, verifiable supervision signal and can be viewed as a step-by-step reasoning trace.
>
> > ***Q4: Finally, would outcome-based RL (as tested here) versus process-based RL that rewards intermediate steps show different results for length scaling, given that shortest-path generation resembles step-by-step reasoning?***
>
> We experimented with simple process-based rewards (e.g., adding length rewards), but they often led to reward hacking (e.g., collapsing to short yet invalid paths to exploit the shaping term) or unstable training dynamics. This phenomenon has also been reported in several prior works [6,7], which attribute such failures to the use of proxy rewards that do not reflect the true task objective and show that verification-based rewards directly encode the goal and are therefore less vulnerable to reward hacking. Since outcome-based RL is the standard setup in recent RL-for-reasoning, we focus on this setting.
>
> ---
> ***References:***
>
> *[1] Swamy, G., Choudhury, S., Sun, W., Wu, Z. S., & Bagnell, J. A. (2025). All roads lead to likelihood: The value of reinforcement learning in fine-tuning.*
>
> *[2] Ma, L., Liang, H., Qiang, M., Tang, L., Ma, X., Wong, Z. H., ... & Zhang, W. (2025). Learning What Reinforcement Learning Can't: Interleaved Online Fine-Tuning for Hardest Questions.*
>
> *[3] Yue, Y., Chen, Z., Lu, R., Zhao, A., Wang, Z., Song, S., & Huang, G. (2025). Does reinforcement learning really incentivize reasoning capacity in llms beyond the base model?*
>
> *[4] Chen, Z., Deng, Y., Yuan, H., Ji, K., & Gu, Q. (2024). Self-play fine-tuning converts weak language models to strong language models.*
>
> *[5] Wu, Y., Zhou, Y., Ziheng, Z., Peng, Y., Ye, X., Hu, X., ... & Yang, X. (2025). On the generalization of sft: A reinforcement learning perspective with reward rectification.*
>
> *[6] Perez, E., Ringer, S., Lukosiute, K., Nguyen, K., Chen, E., Heiner, S., ... & Kaplan, J. (2023, July). Discovering language model behaviors with model-written evaluations.*
>
> *[7] Tarek, M. F. B., & Beheshti, R. (2025). Reward hacking mitigation using verifiable composite rewards*

---

### Author Response · Authors · 2025-12-01
**Final Author Summary: Reviewer Concerns & Corresponding Experiments and Revisions**

We sincerely thank Reviewers UaSt, 7hks, and QLfE for their thoughtful and constructive feedback.

We are glad that the reviewers found the paper **well written** and **easy to understand** (UaSt, QLfE), and highlighted the **fundamental importance** of the generalization questions we study (QLfE). Across reviews, there is consistent recognition that our testbed is **cleverly** constructed (UaSt), enabling a **clear decomposition** of extrapolative generalization (7hks) with a **clean and controlled separation** of its underlying **factors** (UaSt, 7hks). Reviewers further emphasized that our study is **systematic** (UaSt, 7hks), with **well-controlled comparisons** (UaSt) and **detailed, thorough evaluation** (QLfE, UaSt). Finally, they found that the resulting findings provide **principled data-design insights** and **concrete practical takeaways** (UaSt, 7hks), and are **of interest to the broader community** (QLfE).

Although Reviewer UaSt ultimately recommended rejection, their written assessment is nonetheless positive and explicitly notes that the main concern lies in generalizability. We understand that synthetic testbeds with small train-from-scratch models naturally raise questions about external validity. Our new 7B-scale experiment in the math domain, which replicates the key data-design conclusions, can substantially alleviate these concerns.

---

Below we summarize the main reviewer concerns and what we added in the rebuttal:
| Reviewers              | Main Concerns (brief)                                                                                 | What we did in rebuttal (new experiments / analyses)                                                                                                                                                                                                                                                                     |
|------------------------|-------------------------------------------------------------------------------------------------------------|---------------------------------------------------------------------------------------------------------------------------------------------------------------------------------------------------------------------------------------------------------------------------------------------------------------------------|
| UaSt, 7hks, QLfE   | How results from the synthetic navigation domain and small-scale models generalize to other domains/scales | We **added a math-domain experiment** using the **Qwen2.5-7B-Instruct** model (in Section 3.4; Appendix D–E). Building on the existing MathQA dataset, we generated high-quality chain-of-thought solutions using DeepSeek-R1, and defined math-domain analogs of coverage (distinct operation-sets) and diversity (different instantiations/orderings). **Across three MathQA categories (probability/gain/physics)**, we show that training on more distinct questions consistently outperforms adding more solutions per question, and that broader coverage of operation-sets outperforms higher diversity, **mirroring findings from our synthetic setup**. |
| UaSt, 7hks         | Interpretation of why RL does not surpass SFT                                                               | We expanded the RL–SFT analysis (in Section 5) by linking it to the **generation–verification gap**, emphasizing that in our task the optimal path can be computed explicitly, making generation essentially as easy as verification and thus closing the gap where RL typically helps. We also added a detailed **error-type analysis** (in Appendix F), showing *SFT and RL share the same failure modes with nearly identical distributions*, indicating RL introduces no additional capability and cannot bypass SFT errors. |
| QLfE               | Suggestion for adding qualitative examples or failure-case visualizations to further strengthen the paper.                                  | We added qualitative **trajectory visualizations** and a detailed **failure-mode breakdown** for both SFT and RL across different regimes (in Appendix F), providing the interpretability and examples suggested by the reviewer.                                                                                                                                      |

---

> ### Author Response · Authors · 2025-12-01
>
> For completeness and to facilitate the AC’s review, we summarize the revisions made to the manuscript below:
>
> * Added a math-domain case study in **Section 3.4 and Appendix D**, including the prompts used to construct samples with different coverage, diversity, and solutions, as well as a discussion of applicability beyond MathQA in **Appendix E**. (To the practical implication concerns from UaSt, 7hks, QLfE)
> * Expanded the interpretation of why RL does not exceed SFT in **Section 5 (Lines 484–507)**. (UaSt, 7hks)
> * Added full pretraining specifications in **Appendix C.1**. (Q1, 7hks)
> * Added path-length distribution experiments under varying coverage values in **Appendix C.4**. (Q3, 7hks)
> * Added relaxed-metric evaluation, valid-path rate, in **Appendix C.5**. (Q5, 7hks)
> * Added qualitative results and failure-case visualizations in **Appendix F**. (W2 and Q2, QLfE)
>
> ---
>
> *We understand that this year’s ICLR process has been unusually challenging, and we are grateful that you took the time to review our rebuttal and additional experiments. Because we added many fundamentally new experiments, which required substantial effort, our rebuttal was submitted just one day before the unexpected early closure of the discussion phase (about one week before the originally scheduled end). Unfortunately, this left limited time to confirm with the reviewers whether their concerns were fully resolved.*
>
> *However, as the **reviewers’ comments were broadly positive overall and their main substantive concern was the practical implication of the results**, we hope the AC can consider, on their behalf, whether the new additions meaningfully address their concerns.*

---

### Meta-Review · Area_Chair_g5Tj · 2025-12-19

**Summary:**

The paper investigates extrapolative problem solving using a controlled map-navigation testbed, separating spatial transfer from length scaling. The authors demonstrate that transfer is driven by maximising distinct questions (coverage) rather than diversity, while scaling requires neighbouring-length examples. Furthermore, they find that Reinforcement Learning (RL) stabilises optimisation but does not surpass Supervised Fine-Tuning (SFT).

The authors provided a strong rebuttal that addressed concerns about the synthetic domain by replicating their main findings (coverage > diversity) on the MathQA dataset using a Large Language Model (LLM). They also clarified the RL limitations through the generation-verification gap and validated their scaling claims with additional control experiments.

Therefore, I recommend acceptance as a poster.

**Reviewer Concerns:**

1. The most substantial concern---reported by **Reviewers UaSt, 7hks, and QLfE**---was whether findings from a synthetic grid-world navigation task could apply to real-world LLMs. To rebut this, the authors introduced a new experiment using the Qwen2.5-7B-Instruct model on the MathQA dataset, mapping their navigation concepts to mathematical ones, where, for instance, they defined coverage as distinct operation sets. The new experiment replicated their original trade-offs: "High Coverage" consistently outperformed "High Diversity" even in this semantic domain.
2. **Reviewers UaSt and 7hks** questioned the paper's finding that RL did not outperform SFT, noting that this contradicts recent RL for reasoning success stories. The authors argued that RL typically excels when checking a solution is easier than generating it, allowing models to self-correct; however, in their shortest-path task, the optimal path is already known and verifiable during SFT. Consequently, SFT leaves no gap for RL to exploit. The authors supported this claim with a new error analysis showing that their SFT and RL models shared identical failure modes. While they did not improve the RL performance, they turned this contradiction into an insight about the conditions required for RL to add value.
3. **Reviewer 7hks** pointed out missing baselines, requesting details on pre-training convergence, path-length histograms, and long-to-short generalisation tests. The authors clarified that pre-training used 10M trajectories to reach performance saturation and provided histograms (Appendix C.4) to prove that path-length distributions remained stable. They ran the requested long-to-short experiment and achieved >95% success rates. This supports the claim that the failure to scale from short to long paths is due to extrapolation difficulty and not to a failure to learn basic navigation rules.
4. Finally, **Reviewer QYZU** did not provide any substantiated or constructive criticism, and their review has been ignored in my assessment.

**Reviewer Scores:**

While the initial rebuttal was provided early on in the discussion phase, the new results were posted after the OpenReview accident, and none of the Reviewers could comment on them.
Given the lack of interaction, this prospect on score updates is purely speculative:

- **Reviewer UaSt 2->6:** The reviewer's primary reason for rejection was the lack of generalizability to real LLMs. The authors addressed this with the new MathQA experiment using the Qwen2.5-7B-Instruct model, replicating the main findings.

- **Reviewer 7hks 6->7/8:** The reviewer was already supportive but requested specific technical controls. The authors provided all the controls confirming the findings.

- **Reviewer QLfE 6->7:** This reviewer criticized the "toy" nature of the domain and requested qualitative visualizations. Given the new experiment on MathQA and the new visualizations in Appendix F, I would expect a score increase.

- **Reviewer QYZU 2->2:** The critique focused on *conceptually incoherent* presentation and a specific analogy in the abstract. As the authors focused their rebuttal on technical concerns, this reviewer's score likely remains unchanged.

---

### Decision · Program_Chairs · 2026-01-26

Accept (Poster)